# LEMON: LOSSLESS MODEL EXPANSION

**Yite Wang**[1,*], **Jiahao Su**[2,†], **Hanlin Lu**[2], **Cong Xie**[2], **Tianyi Liu**[2], **Jianbo Yuan**[2],
**Haibin Lin**[2], **Ruoyu Sun**[3,4], **Hongxia Yang**[2]
[1]University of Illinois Urbana-Champaign, USA  [2]ByteDance Inc.
[3]The Chinese University of Hong Kong, Shenzhen, China [4]Shenzhen Research Institute of Big Data
`yitew2@illinois.edu` `{jiahao.su, hanlin.lu, cong.xie, tianyi.liu,`
`jianbo.yuan, haibin.lin, hx.yang}@bytedance.com` `sunruoyu@cuhk.edu.cn`

## ABSTRACT

Scaling of deep neural networks, especially Transformers, is pivotal for their surging performance and has further led to the emergence of sophisticated reasoning capabilities in foundation models. Such scaling generally requires training large models from scratch with random initialization, failing to leverage the knowledge acquired by their smaller counterparts, which are already resource-intensive to obtain. To tackle this inefficiency, we present **L**ossl**E**ss **MO**del Expansio**N** (LEMON), a recipe to initialize scaled models using the weights of their smaller but pre-trained counterparts. This is followed by model training with an optimized learning rate scheduler tailored explicitly for the scaled models, substantially reducing the training time compared to training from scratch. Notably, LEMON is versatile, ensuring compatibility with various network structures, including models like Vision Transformers and BERT. Our empirical results demonstrate that LEMON reduces computational costs by 56.7% for Vision Transformers and 33.2% for BERT when compared to training from scratch.

## 1 INTRODUCTION

Deep neural networks (DNNs) have become increasingly popular, showcasing their adaptability across natural language processing (Liu & Lapata, 2019; Achiam et al., 2023), computer vision (Chen et al., 2023a;b), and code generation (Yu et al., 2023). Recent advances in architectural design, especially Transformers, have further enhanced the scalability of DNNs. However, it is a common practice to train large-scaled models from scratch, discarding the learned knowledge in their smaller counterparts. Such an approach can be highly inefficient, especially given the intensive computational resources required to train large language models such as Generative Pre-trained Transformer (GPT) (Brown et al., 2020), and the resultant huge carbon footprints. For instance, training GPT-3 incurs costs around $4.6M (Li, 2020). Given these challenges, researchers are keenly exploring ways to leverage the prior knowledge of smaller models for more efficient scaling.

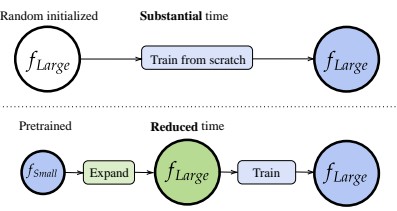

Figure 1: Comparison between training from scratch and model expansion. In model expansion, a smaller pre-trained model is expanded to a larger model without any performance drop, requiring significantly less training time than training from scratch.

Knowledge inheritance and model expansion are two primary methodologies to achieve this goal. Knowledge inheritance (Qin et al., 2021), the reverse of knowledge distillation (Hinton et al., 2015), allows the large model to learn the predictions of a smaller pre-trained model. However, this method often necessitates additional computational resources and modifications to the training pipeline due to the involvement of a 'teacher network.' In contrast, model expansion directly utilizes the weights from the pre-trained small source network, either without training (Chen et al., 2015; 2021a; Yang et al., 2020; Shen et al., 2022) or with negligible training (Wang et al., 2023a). Hence, our work

---
*Work done during internship at ByteDance.
†Corresponding author.

mainly focuses on model expansion due to its minimal impact on the training pipeline and negligible computational overhead.

A compelling requirement for model expansion is to ensure it is 'lossless,' meaning no information from the source model is lost. Specifically, the goal is for the larger target model to inherit the exact functional mapping as the smaller source model, thus preserving the performance. Net2Net (Chen et al., 2015) represents a foundational study of lossless model expansion for convolutional networks (CNNs) and multi-layer perceptrons (MLPs) where it duplicates neurons and averages their fan-out weights. However, a challenge arises with the 'weight symmetry' issue. This problem occurs when duplicated neurons in expanded layers introduce redundancy, which persists during subsequent training. In this sense, the expanded model will never gain more capacity than the source model. To counteract this problem, previous researchers introduced additional noise into the expansion process, leading to a shift away from a genuine lossless expansion.

Transformers, despite their rising popularity in modern deep learning, introduce additional complexities in achieving lossless expansion that goes beyond traditional issues like weight symmetry. One key obstacle arises from the intricacy of the LayerNorm, which was evident when bert2BERT (Chen et al., 2021a) tried extending the Net2Net approach to Transformers, leading to lossy outcomes. Staged training (Shen et al., 2022) demonstrated the feasibility of lossless model expansion, but with a specific constraint: doubling the width during expansion and only for a variant of Transformers known as Pre-Layer Normalization (Pre-LN) Transformers. However, real-world applications often require width increases in the expanded model that are indivisible by the smaller source model's width, highlighting a limitation in existing methodologies. A typical scenario involves expanding the hidden dimension from 512 to 768.

In exploring the possibilities of lossless model expansion, our research focuses on the ability to **break weight symmetry**, handle **indivisible width and depth increments**, and remain compatible with almost **all Transformer varieties**. We have discovered affirmative answers, revealing that multiple solutions exist, enabling the selection of an optimal candidate to break the weight symmetry or find an initialization point with specific properties. Specifically, we break the weight symmetry of replicated neurons by setting their fan-out weights to be unequal, and we introduce average expansion to deal with LayerNorm for indivisible width increment.

In addition to lossless model expansion techniques, our study also delves into training recipes for the expanded models. It is often overlooked whether applying the original training recipe remains optimal or whether the expanded models necessitate tailored approaches. Our empirical studies reveal two key insights: expanded models can benefit from utilizing a default maximum learning rate and, intriguingly, a learning rate scheduler that decays more rapidly.

Our contributions are summarized as follows:

1. We propose LEMON, a suite of algorithms designed for lossless model expansion across a variety of architectures, ensuring compatibility with indivisible width and depth increments.

2. Drawing inspiration from our empirical results, we propose an optimized learning rate scheduler for the expanded models. This scheduler maintains the maximum learning rate used by training from scratch, but features accelerated decay rates.

3. LEMON reduces the computational costs by up to 56.7% for Vision Transformers and 33.2% for BERT compared to training from scratch, thereby setting a new benchmark in performance.

## 2 RELATED WORKS

**From small models to larger models.** There are two main approaches to transferring the knowledge of the smaller models to larger models: knowledge inheritance and model expansion. Knowledge inheritance (Qin et al., 2021) enables a student network to learn the logits provided by a teacher network. Net2Net (Chen et al., 2015) was the first work to explore the idea of model expansion. It involves randomly duplicating neurons while preserving the output values through proper normalization and increasing depth by adding identity layers. However, Net2Net resorts to introducing weight perturbations to overcome weight symmetry, resulting in performance deterioration. Follow-up work bert2BERT (Chen et al., 2021a) extends Net2Net to Transformer while others study depth growth (Gong et al., 2019; Yang et al., 2020; Chang et al., 2017; Dong et al., 2020). Staged training

Table 1: Overview of model expansion or knowledge inheritance methods. In the first three columns, we use symbols ✓, ✗, and N/A to denote whether the method is (1) lossless, (2) non-lossless, or (3) not applicable in the given scenarios. Here, 'Depth' represents the scenario where the large model has more layers than the smaller model, and 'Width (divisible/indivisible)' denotes whether the large model's hidden dimension is a multiple of the smaller model's. In the subsequent columns, 'Non-unique Expansion' denotes whether the expansion is unique (e.g., produce target models to break weight symmetry). 'Data-free' specifies whether the algorithm requires training data. LEMON is the most versatile method compared to previous methods.

| Method | Depth | Width (divisible) | Width (indivisible) | Non-unique Expansion | Data-free |
|---|---|---|---|---|---|
| KI Qin et al. (2021) | ✗ | ✗ | ✗ | No | No |
| StackBERT (Gong et al., 2019) | ✗ | N/A | N/A | No | Yes |
| MSLT (Yang et al., 2020) | ✗ | N/A | N/A | No | Yes |
| bert2BERT (Chen et al., 2021a) | ✗ | ✓ | ✗ | No | Yes |
| Staged Training (Shen et al., 2022) | ✓ | ✓ | N/A | No | Yes |
| LiGO (Wang et al., 2023a) | ✗ | ✗ | ✗ | Yes | No |
| **LEMON (Ours)** | ✓ | ✓ | ✓ | Yes | Yes |

(Shen et al., 2022) made significant progress by proposing a lossless model expansion method for Pre-LN Transformer, but with the constraint of width doubling. LiGO (Wang et al., 2023a) suggests employing multiple training steps to find an appropriate linear combination of weights from the source networks. Despite these advancements, all existing methods still face the challenge of the performance drop or strict restrictions on the model width. Table 1 compares the related methods.

**Network initialization.** Numerous studies aim to seek optimal initialization methods for neural networks, primarily focusing on regulating the norm of network parameters (Glorot & Bengio, 2010; He et al., 2015). Theoretical works try to study these methods through dynamical isometry (Saxe et al., 2013) or mean field theory (Poole et al., 2016). Orthogonal initialization, which supports layer-wise dynamical isometry in fully-connected layers, has been extended to CNNs via Delta orthogonal initialization (Xiao et al., 2018). However, there has been limited research on initialization methods specifically for Transformers. Most of these works focus on theoretical approaches to train Transformers without skip connections or normalization layers (Noci et al., 2022; He et al., 2023). Mimetic initialization (Trockman & Kolter, 2023) seeks to initialize attention based on the principles of pre-trained Transformers.

**Continual pre-training.** Recent research explores adapting pre-trained networks for new or improved datasets. While some target datasets from different domains (Scialom et al., 2022; Ke et al., 2022; Gupta et al., 2023; Qin et al., 2022), others focus on datasets that evolve over time (Han et al., 2020; Jang et al., 2021; Loureiro et al., 2022). Model expansion is similar to continual pre-training, with the distinction being a change in the model size rather than the data distribution.

## 3 PRELIMINARIES

**Model expansion** aims to initialize a large model with the weights from its smaller pre-trained counterparts. Concretely, suppose we have pre-trained weights $\theta_S$ in a source network $f_S(\cdot; \theta_S^{\text{trained}})$, our goal is to design a mapping $\theta_T^{\text{expanded}} = \mathcal{M}(\theta_S^{\text{trained}})$, where the expanded weights initialize the target network as $f_T(\cdot; \theta_T^{\text{expanded}})$. Since these expanded weights contain knowledge acquired by the small pre-trained model, it should accelerate the training of $f_T$ compared to random initialization. Moreover, we call a model expansion algorithm **lossless** if $f_T(\mathbf{x}; \theta_T^{\text{expanded}}) = f_S(\mathbf{x}; \theta_S^{\text{trained}}), \forall \mathbf{x}$.

An example for model expansion is to use a pre-trained ResNet-50 (He et al., 2016) or BERT-Small ($f_S$) to facilitate the training of WideResNet-110 or BERT-Base ($f_T$), respectively. Instead of training the larger models from scratch, the idea is to initialize them with the weights of their smaller pre-trained counterparts, i.e., ResNet-50 or BERT-Small.

**Transformer architecture,** introduced by Vaswani et al. (2017), consists of multiple Transformer blocks $g(\cdot)$, where each block is a stack of two modules, a multi-head attention (MHA) and a two-layer MLP. Depending on the location of LayerNorm, Trans-

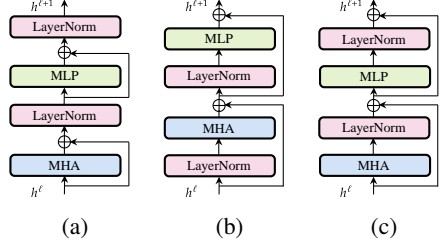

Figure 2: Varieties of attention blocks. **(a)** Post-LN block. **(b)** Pre-LN block. **(c)** Res-Post-Norm block.

former blocks can be categorized as (1) Post-LN block used by the original BERT (Devlin et al., 2019) where `LN` is applied after the residual block, i.e., $g(x) = \texttt{LN}(\texttt{Module}(x) + x)$, (2) Pre-LN used by GPT (Brown et al., 2020), Pre-LN BERT, Vision Transformers (Dosovitskiy et al., 2021), and SWin Transformer (Liu et al., 2021b) where `LN` is applied inside the residual connection and before all other transformations, i.e., $g(x) = x + \texttt{Module}(\texttt{LN}(x))$, and (3) Res-Post-Norm used by SWin Transformer V2 (Liu et al., 2022) where `LN` is applied inside the residual connection and after all other transformations, i.e., $g(x) = x + \texttt{LN}(\texttt{Module}(x))$. See Figure 2 for an illustration.

**Multi-head attention (MHA)** uses multiple self-attention heads to attend to information from different representation subspaces of the input. Given an input sequence $\mathbf{X} \in \mathbb{R}^{E \times D}$, where $E$ is the sequence length, and $D$ is the embedding dimension, each head projects the inputs into different subspaces using linear transformations. For the $i$-th head, its query is defined as $\mathbf{Q}_i = \mathbf{X}\mathbf{W}_i^Q$, its key as $\mathbf{K}_i = \mathbf{X}\mathbf{W}_i^K$, and its values as $\mathbf{V}_i = \mathbf{X}\mathbf{W}_i^V$, where $\mathbf{W}_i^Q, \mathbf{W}_i^K \in \mathbb{R}^{D \times d_K}$ and $\mathbf{W}_i^V \in \mathbb{R}^{D \times d_V}$. Here, $d_K$ and $d_V$ represent the dimensions of the key and value, respectively. Each head then computes the attention with $\text{Head}_i = \text{Attention}(\mathbf{Q}_i, \mathbf{K}_i, \mathbf{V}_i) = \texttt{softmax}\left(\mathbf{Q}_i\mathbf{K}_i^\mathsf{T}/\sqrt{d_K}\right)\mathbf{V}_i$. The outputs from all $H$ heads are concatenated and linearly transformed to yield the final output:

$$\text{MHA}(\mathbf{X}) = \texttt{Concat}\left[\text{head}_1, \cdots, \text{head}_H\right]\mathbf{W}^O,$$

where $\mathbf{W}^O \in \mathbb{R}^{Hd_V \times D}$ is the weight matrix. Please refer to Vaswani et al. (2017) for more details.

**Weight symmetry.** Consider a two-layer MLP with two hidden neurons in the form of $\texttt{MLP}(\mathbf{x}) = \mathbf{v}^\mathsf{T}\sigma(\mathbf{W}_1\mathbf{x}) = v_1\sigma(w_{1,1}x_1 + w_{1,2}x_2) + v_2\sigma(w_{2,1}x_1 + w_{2,2}x_2)$, where $\sigma$ is the nonlinear activation, and $v_1, v_2$ are the weights associated with the hidden neurons. If the weights are initialized such that $v_1 = v_2, w_{1,1} = w_{2,1}, w_{1,2} = w_{2,2}$, the two neurons will always compute identical values throughout training. This symmetry results from the fact that, at each iteration, the gradients for the corresponding weights are the same, i.e., $\dot{w}_{1,1} = \dot{w}_{2,1}, \dot{w}_{1,2} = \dot{w}_{2,2}$. Weight symmetry is detrimental as it implies that the two symmetric neurons do not contribute independently to the model's learning, potentially harming the model's expressive power and learning capability.

## 4 LOSSLESS MODEL EXPANSION

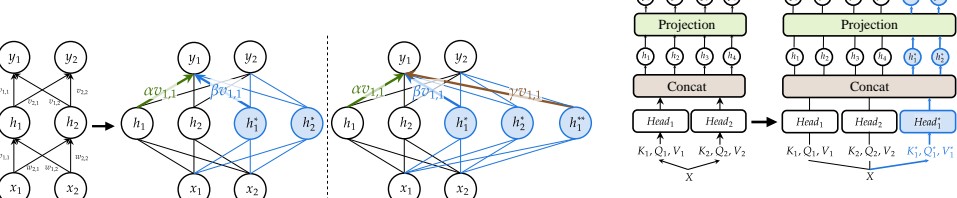

(a) Width expansion of MLP from 2 to 4 (**left**) or 5 (**right**).

(b) Expand the number of heads in MHA from 2 to 3.

Figure 3: Lossless width expansion with weight symmetry breaking for multi-layer perceptron (MLP) and multi-head attention (MHA). **(a) Left**: MLP expansion with divisible width. We replicate neurons $h_1/h_2$ to $h_1^*/h_2^*$ and set $\alpha + \beta = 1$ with $\alpha \neq \beta$. **Right**: MLP expansion with indivisible width. We further replicate the neuron $h_1$ to $h_1^{**}$ and set $\alpha + \beta + \gamma = 1$ with $\alpha \neq \beta \neq \gamma$. **(b)** MHA expansion with head dimension unchanged. We duplicate $\text{Head}_1$ to $\text{Head}_1^*$ (i.e., duplicate key/query/value projections) and expand the projection layer as in an MLP module.

We decompose the expansion operator $\mathcal{M}$ to two operators, i.e. the depth expansion operator $\mathcal{D}$ and the width expansion operator $\mathcal{W}$, each applied to individual layers.

Our expansion method mainly consists of three main components, i.e., (1) general lossless width expansion with symmetry breaking, (2) average width expansion for LayerNorm, and (3) lossless depth expansion. In the expansion process, each layer is independently subjected to these methods, ensuring a layer-level lossless expansion. This entails a systematic, recursive application of duplicating inputs for each layer in a lossless manner, and every layer, in turn, guarantees the lossless duplication of its output.

### 4.1 GENERAL LOSSLESS WIDTH EXPANSION WITH SYMMETRY BREAKING

We first show how to apply lossless expansion with symmetry breaking for (1) fully-connected layers (FC-layers) and (2) multi-head attention (MHA).

**Lossless width expansion for FC-layers.** Transformers consist of a set of FC-layers. We first use MLP as an example to show the basic width expansion operator for the FC-layers.

For width expansion, we create copies of neurons similar to Net2Net and bert2BERT, as this step is necessary due to the nonlinear activation used in MLP. However, the essential difference is that we do **NOT** set the fan-out weights of replicated neurons to be equal. Out of simplicity, we use a single-hidden-layer MLP for illustration, and we show it on the left half in Figure 3a . We first replicate neurons $h_1, h_2$ to $h_1^*, h_2^*$ in a circular pattern. Consider the same neurons $h_1$ and $h_1^*$ in the plot with the original fan-out weight $v_{1,1}$; we can set the expanded fan-out weights to be $\alpha v_{1,1}$ and $\beta v_{1,1}$ where $\alpha + \beta = 1$ to ensure lossless expansion.

The selection of $(\alpha, \beta)$ corresponds to a specific lossless model expansion algorithm, and our method can be considered as a generalization of existing model expansion methods. Specifically, Net2Net and bert2BERT perform width expansion by setting $\alpha = \beta = 1/2$. However, such a choice causes weight symmetry problems where two neurons learn the exact same representations when it is initialized and for the subsequent training. We introduce a simple modification to fix the issue, i.e., by setting $\alpha \neq \beta$ is enough to break weight symmetry for commonly-used nonlinear activation $\sigma$. This concept extends to cases where neurons are replicated more than twice, illustrated on the right half of Figure 3a. In such cases, we set coefficients such that $\alpha + \beta + \gamma = 1$ and $\alpha \neq \beta \neq \gamma$.

**MHA expansion.** We make sure that we directly copy the entire head in a circular pattern similar to FC-layers as mentioned in the previous section. We then perform width expansion for the corresponding key, query, and value matrices. Then, it reduces to a case similar to MLP due to the following projection matrix. Symmetry breaking is realized by setting the corresponding fan-out weights in the projection matrix differently. We illustrate the process in Figure 3b.

### 4.2 AVERAGE WIDTH EXPANSION FOR LAYERNORM

When dealing with indivisible width increments, we need to design a specific expansion method for the LayerNorm layer. In this section, we demonstrate that achieving a lossless expansion is feasible provided that FC-layers are positioned before the LayerNorm layer.

**Average width expansion.** We first show that it is easy to perform the average expansion method such that the output of FC-layers is padded with its average. We do so by adding neurons whose weights are the average of existing neurons. Specifically, we pad the original weight $\mathbf{W} \in \mathbb{R}^{D_{out} \times D_{in}}$ with rows $1/D_{out} \sum_i^{D_{out}} \mathbf{W}[i]$, and pad bias $\mathbf{b} \in \mathbb{R}^{D_{out}}$ with $1/D_{out} \sum_i^{D_{out}} \mathbf{b}[i]$.[1] See Figure 4 for an illustration.

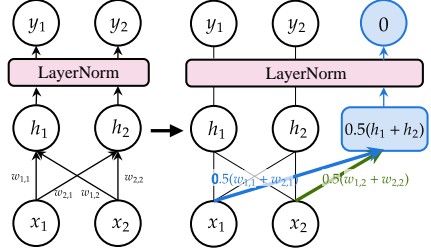

Figure 4: Lossless average expansion. When the fully-connected layer right before Layer-Norm is average expanded, the output of LayerNorm is expanded with zeros.

**LayerNorm layer.** We now show that if the input of LayerNorm is average expanded, lossless width expansion is possible. Specifically, consider LayerNorm layer with element-wise affine-transformation in the form of $\mathtt{LN}(\cdot; \boldsymbol{\mu}, \mathbf{b}) = \boldsymbol{\mu} \odot \mathtt{Norm}(\cdot) + \mathbf{b}$, where $\boldsymbol{\mu}, \mathbf{b} \in \mathbb{R}^{D_S}$ and $D_T \leq 2D_S$. Define average expanded of $\mathbf{x} \in \mathbb{R}^{D_S}$ to be $\mathbf{x}^* \in \mathbb{R}^{D_T}$. It can be shown that $\mathtt{LN}(\mathbf{x}^*; \boldsymbol{\mu}^*, \mathbf{b}^*) = \mathtt{Concat}\left[\mathtt{LN}(\mathbf{x}; \boldsymbol{\mu}, \mathbf{b}), \mathbf{0}\right]$ if $\boldsymbol{\mu}^* = \mathtt{Concat}\left[\eta \boldsymbol{\mu}, \boldsymbol{\zeta}\right]$ and $\mathbf{b}^* = \mathtt{Concat}\left[\mathbf{b}, \mathbf{0}\right]$, where $\mathbf{0} \in \mathbb{R}^{D_T - D_S}$ is a zero vector, $\boldsymbol{\zeta} \in \mathbb{R}^{D_T - D_S}$ is an arbitrary vector, and $\eta = \sqrt{(D_S/D_T)}$ is a scalar. See section E.1 for results and proof with a more generalized case where $D_T \geq D_S$.

### 4.3 LOSSLESS DEPTH EXPANSION

In this section, we detail our approach for increasing model depth in a lossless manner.

---

[1]Input dimension should be expanded as well depending on how inputs are expanded.

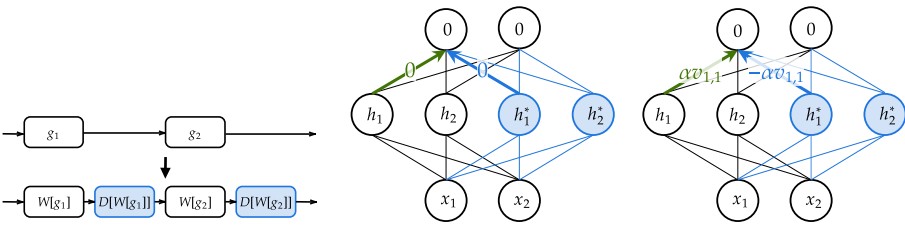

(a) Arrangement of block stacking.   (b) Type-1 depth expansion.   (c) Type-2 depth expansion.

Figure 5: Lossless depth expansion. **(a)** We place a new block next to the block where it originates. **(b)** For type-1 depth expansion, we set the weights of the last fully-connected layer to zeros. **(c)** For type-2 depth expansion, we specify the weights of the last fully-connected layer so that the contributions from replicated neurons cancel each other. For example, assume $h_1^*$ is a duplicate of $h_1$, we set their fan-out weights to be $\alpha v_{1,1}$ and $-\alpha v_{1,1}$ to enforce zero output.

**Arrangement of added layers.**   Similar to how Chang et al. (2017); Dong et al. (2020) deal with ResNet, we put added layers directly next to the source layer. For example, when expanding two-layer network with blocks $\{g_1, g_2\}$, we perform depth expansion with the resulting model $\{\mathcal{W}[g_1], \mathcal{D}[\mathcal{W}[g_1]], \mathcal{W}[g_2], \mathcal{D}[\mathcal{W}[g_2]]\}$. See Figure 5a for an illustration.

**Lossless depth expansion.**   We now provide two ways to perform lossless depth expansion. Firstly, we can simply set the output of each module (MLP or MHA) to be zero, i.e. $\alpha = \beta = 0$. Hence, the residual branch does not contribute to the output. This choice gives great flexibility to the rest of the parameters since we can (1) copy weights from other layers or (2) randomly initialize the weights. See Figure 5b for an illustration. Secondly, we can enforce the output to be zero by setting the summation of fan-out weights for replicated neurons to zero. With the example shown in Figure 3a, we can set the fan-out weights of replicated neurons to be $\alpha = -\beta \neq 0$ to ensure all outputs are zeros.[2] See Figure 5c for an illustration.

## 4.4   A SUMMARY OF IMPLEMENTING MODEL EXPANSION

We summarize the procedure of model expansion for Pre-LN Transformer architecture with both depth and width increments. We first average expand the embedding weights. Then, make sure the output of each layer is average expanded. Hence, the input to the decoder layer is the original output padded with zeros after the last LayerNorm. We provide a detailed description of our expansion method in section C.1. Furthermore, we explain how to use our method for Post-LN and Res-Post-Norm architectures in Appendix D.

## 5   HOW TO TRAIN THE EXPANDED MODELS

In this section, we delve into the influence of different factors in the training recipe, in particular the maximum learning rate and the learning rate scheduler, when training expanded models.

**Experiment setup.**   Throughout this study, we adopt ViT (Dosovitskiy et al., 2021) as our exemplary model and train it on the standard ImageNet-1k dataset. In particular, we choose to expand $\text{ViT}(6, 512)$ to $\text{ViT}(12, 768)$, where $6/12$ represent the number of attention blocks and $512/768$ denote the hidden dimensions. When training these models from scratch, we apply a default maximum learning rate of $1 \times 10^{-3}$ and run the training for 300 epochs with a batch size of 1024. We use a cosine learning rate scheduler that decays to a minimum learning rate of $10^{-5}$. However, we will modify this training recipe for continual training of the expanded model $\text{ViT}(12, 768)$.

### 5.1   THE EFFECTS OF MAXIMUM LEARNING RATE

Suppose we have an expanded model, $f_T$, that maintains the same accuracy as a smaller source model, $\mathcal{A}(f_S)$. One might naturally opt for a smaller learning rate, expecting the validation accuracy of the expanded model to continue to decrease. If this were the case, we could smooth the

---

[2]If neurons are not replicated, then we have to set the fan-out weights to be zero.

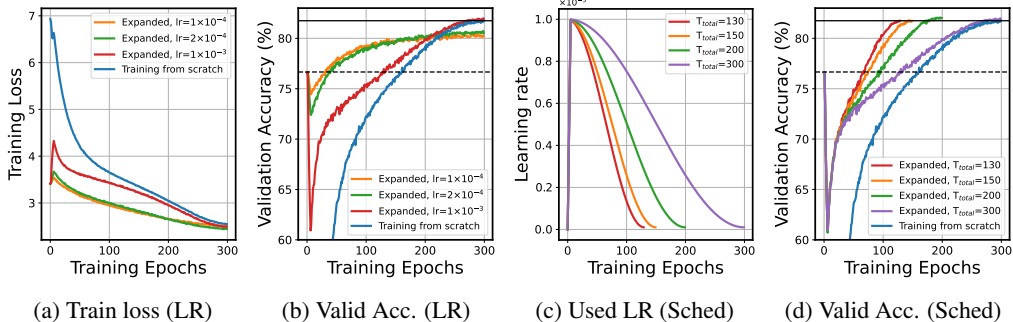

| (a) Train loss (LR) | (b) Valid Acc. (LR) | (c) Used LR (Sched) | (d) Valid Acc. (Sched) |

Figure 6: Influence of maximum learning rate **(LR; a,b)** and learning rate scheduler **(Sched; c,d)** for training expanded Vision Transformers. Dashed and solid horizontal lines represent the validation accuracy of small and large models, when trained from scratch. **(a)** Train loss when changing maximum LR, **(b)** validation accuracy when changing maximum LR, **(c)** different LR scheduler used in experiments, **(d)** validation accuracy when changing LR scheduler. We find that (1) using a smaller maximum LR results in smaller training loss but yields worse validation accuracy; (2) expanded models require significantly fewer epochs to match the performance of the larger model.

transition between the training processes of the small model and the expanded model. However, our investigations reveal that the relationship is more complex than it initially seems.

We conducted experiments with three different maximum learning rates: $1 \times 10^{-3}$ (default), $2 \times 10^{-4}$, and $1 \times 10^{-4}$, maintaining a consistent minimum learning rate of $1 \times 10^{-5}$ across all cases. The results are shown in Figure 6b. We summarize our findings in the following paragraphs.

**Performance drop early at training.** An interesting observation is the immediate decrease in validation accuracy experienced by all three expanded models early during the learning rate warm-up.[3] This performance drop is correlated with the magnitude of the learning rate; the larger it is, the more pronounced the drop. This aligns with our anticipation as smaller learning rates are critical for model convergence, especially when the source model is already near local optima. Adopting a larger learning rate can displace the weights from this local minimum, leading to an increase in training loss.

**Maximum learning rate and model generalization.** We observe that maintaining the default maximum learning rate is pivotal to recovering the performance of the large model. To investigate whether adopting smaller learning rates hinders model learning, we also examine the training loss of all cases, as illustrated in Figure 6a. The results show that models trained with reduced learning rates incur smaller training losses compared to training from scratch. Hence, we postulate that the deterioration in performance, induced by a smaller maximum learning rate, is detrimental to the generalization capability of the expanded networks rather than the optimization capability. This concept is also theoretically examined by Li et al. (2020), illustrating how the learning rate can influence the sequence of learning varied patterns, thereby affecting generalization capacities.

## 5.2 How fast the learning rate should decay

After settling the maximum learning rate, the next important parameter to consider is the total number of epochs. Most works use the default learning rate scheduler (Wang et al., 2023a; Chen et al., 2021a), maintaining the same number of epochs as if the model were training from scratch. We, however, note that the expanded model, having inherited knowledge from the source model, starts with a small training loss — this holds true even when accounting for the significant loss drop during warm-up. This indicates the expanded model is closer to the local optimum, requiring a smaller learning rate for continued loss reduction. Thus, we should adopt a learning rate scheduler where the learning rate decays faster.

We examine four different epoch totals $T_{\text{total}}$: 130, 150, 200, and 300, with the corresponding learning rate schedulers illustrated in Figure 6c. Experiment results are shown in Figure 6d.

---

[3]We tried to change the number of warm-up steps, but the results were not greatly affected.

**Expanded model necessitates faster learning rate decay.** As depicted in Figure 6d, a notable observation is that employing a learning rate scheduler with faster decay enables the expanded model to quickly attain the performance of the corresponding large target model. Remarkably, the expanded model requires only 130 epochs of training to match the performance of the target model that was trained from scratch, translating to a computational cost saving of up to $56.67\%$. This corroborates our earlier conjecture that expanded models need a learning rate scheduler that decays faster.

In summary, we recommend employing the same maximum learning rate as is used for training from scratch but with accelerated decay.

## 6 MAIN EXPERIMENTS

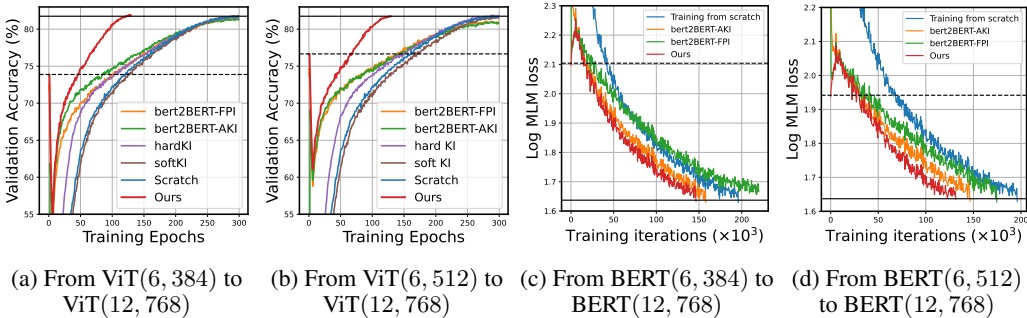

(a) From ViT$(6, 384)$ to ViT$(12, 768)$

(b) From ViT$(6, 512)$ to ViT$(12, 768)$

(c) From BERT$(6, 384)$ to BERT$(12, 768)$

(d) From BERT$(6, 512)$ to BERT$(12, 768)$

Figure 7: Results of ViT on ImageNet **(a,b)** and BERT on English Wiki **(c,d)**. Dashed and solid horizontal lines represent the validation accuracy/MLM loss of the trained small model and target model. LEMON outperforms baselines, yielding computational savings of 56.7%, 56.7%, 25.5%, and 33.2% in panels **(a)**, **(b)**, **(c)**, and **(d)** compared to training from scratch, respectively.

In this section, we compare our method with existing model expansion algorithms on Vision Transformers and BERT. We name our method **L**ossl**E**ss **MO**del Expansio**N** (LEMON), which uses the expansion algorithm explained in section 4 with an optimized learning rate scheduler that decays faster, as suggested in section 5.

**Baselines.** We consider several baselines to compare with our proposed method: (1) training the target model from scratch, (2) bert2BERT-FPI (Chen et al., 2015), a generalization of Net2Net, (3) bert2BERT-AKI (Chen et al., 2021a), which uses advanced knowledge initialization (AKI) to break weight symmetry, (3) soft KI (Qin et al., 2021) which learns the output of the source model by minimizing the KL-divergence of the two distributions, and (4) hard KI which learns the predictions of the source model. We do not include StackBERT (Gong et al., 2019), Yang et al. (2020), and Staged training (Shen et al., 2022) as they are not compatible with indivisible width expansion. LiGO (Wang et al., 2023a) is unavailable for direct comparison due to the absence of open-source code; hence, comparisons are made using reported values on ViT(12,512) to ViT(12,768) in section F.1. Experiments of CNN and Post-LN BERT can be found in section F.2 and section F.3, respectively.

### 6.1 VISION TRANSFORMERS

**Experiment setting.** We adopt the default experimental setup described in section 5 unless stated otherwise. For LEMON, the learning rate is decayed to its minimum value over $T_{\text{total}} = 130$ epochs in both experiments. Parameters choices of LEMON are discussed in section C.4.

**Experiment results.** As demonstrated in Figure 7a and Figure 7b, LEMON is able to achieve lossless model expansion. For both experiment settings, LEMON is able to recover the performance of the target model in 130 epochs, outperforming other baselines.

Several additional observations were made during the study. First, both bert2BERT-FPI and bert2BERT-AKI exhibited performance inferior to training from scratch. Second, consistent with the observations in Chen et al. (2021a) and Wang et al. (2023a), soft KI did not enhance the training speed of the target model, while hard KI did, possibly by functioning akin to curriculum learning and filtering out the challenging training samples for the target model early at training.

Table 2: Downstream performance of BERT$(12, 768)$ on the GLUE dataset: Large model expanded from BERT(6,384) achieves the best downstream performance. A potential reason for outperforming BERT(6,512) may be its longer training duration (165k) compared to the BERT(6,512) (132k).

| Total training steps | Dataset (Metric) | STS-B (Corr.) | MRPC (Acc.) | CoLA (Mcc.) | SST-2 (Acc.) | QNLI (Acc.) | MNLI (Acc.) | MNLI-mm (Acc.) | QQP (Acc.) |
|---|---|---|---|---|---|---|---|---|---|
| 220k | Train from scratch | 0.744 | 83.33 | 0.19 | 88.88 | 87.80 | 80.28 | 81.17 | 89.62 |
| 132k | LEMON (Ours), from BERT$(6, 512)$ | 0.848 | 83.82 | 0.36 | 90.14 | 88.76 | 80.92 | 81.57 | 89.91 |
| 165k | LEMON (Ours), from BERT$(6, 384)$ | **0.866** | **85.54** | **0.38** | **90.94** | **89.33** | **81.81** | **81.81** | **90.40** |

## 6.2 LANGUAGE MODELS

**Experiment setting.** For our experiments, we train Pre-LN BERT (Xiong et al., 2020) on masked language modeling task. The model is trained on the English Wiki corpus as per the methods in Tan & Bansal (2020) for 220k iterations with 5k warmup steps and a batch size of 256. We use a maximum learning rate of $2 \times 10^{-4}$ and a cosine learning rate scheduler which decreases the learning rate to $2 \times 10^{-5}$. Following Liu et al. (2019), we remove the next sentence prediction task and use a fixed sequence length of 128 for model pre-training.

We consider the following expansion procedure: (1) BERT$(6, 384)$ to BERT$(12, 768)$, and (2) BERT$(6, 512)$ to BERT$(12, 768)$. We remove KI as our baseline. For LEMON, we decay the learning rate to the minimum values in 165k and 132k iterations for BERT$(6, 384)$ and BERT$(6, 512)$, respectively. Parameters choices of LEMON are discussed in section C.4. We report the number of iterations needed to achieve a log validation MLM loss of 1.64.

**Experiment results.** As shown in Figure 7c and Figure 7d, LEMON successfully expands smaller models without incurring loss. It outperforms baselines and achieve computational cost savings of 25.5% and 33.2% for BERT$(6, 384)$ and BERT$(6, 512)$, respectively.

**Downstream task.** We also present downstream performance of BERT trained by LEMON on the GLUE (Wang et al., 2018) dataset. We report correlation for the STS-B dataset and Matthews correlation coefficient for the CoLA dataset. Accuracy is reported for the remaining datasets. The results reveal that BERT(12,768) exhibits superior downstream performance when expanded from BERT(6,384) as opposed to being trained from scratch or being expanded from BERT(6,512). This likely stems from its more extensive training duration (165k iterations) compared to the model expanded from BERT(6,512) (132k iterations).

## 6.3 ABLATION STUDIES: THE EFFECTS OF THE TRAINING RECIPE

To study the effects of our proposed training recipe on baselines, we conduct an ablation study where we apply our training recipe on them. The results are shown in Figure 8a. It is shown that expanded models indeed require faster learning rate decay. Additionally, LEMON continues to outperform other baselines under the same modified training recipe.

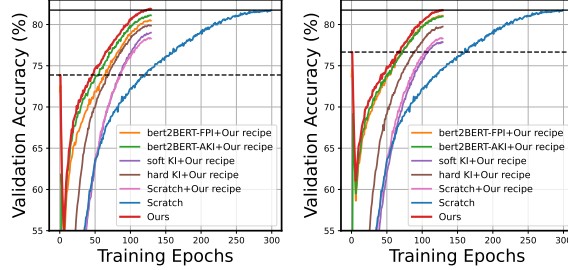

(a) ViT$(6, 384) \rightarrow (12, 768)$. (b) ViT$(6, 512) \rightarrow (12, 768)$.

Figure 8: LEMON outperforms other baselines even when they employ the same optimized learning rate schedulers.

## 7 CONCLUSION

In this paper, we propose LEMON, a method that combines lossless model expansion and optimized learning rate scheduler, showing compatibility and significant performance improvements for a variety of Transformer architectures. However, LEMON does have its limitations, including the need for tuning the total number of training epochs, and our evaluation scale was constrained by available computational resources. Looking ahead, we are working on extending the application of LEMON to larger models and on developing methodologies for selecting optimal free parameters when initializing LEMON.

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

OVERVIEW OF THE APPENDIX

The Appendix is organized as follows:

- Appendix A introduces the general experiment setup.
- Appendix B provides backgrounds and notations for model expansion.
- Appendix C shows details for applying LEMON on Pre-LN Transformers.
- Appendix D shows details for applying LEMON on other architectures.
- Appendix E provides related proofs.
- Appendix F provides additional experiments.
- Appendix G provides additional related works for efficient deep learning.

## A  EXPERIMENT SETUP

We conduct all experiments with NVIDIA-V100 and NVIDIA-A100 GPUs. We use the official code base of DeiT[4] (Touvron et al., 2021) for training Vision Transformers and the code base of VLM[5] (Tan & Bansal, 2020) for training BERT. For CNN experiments, we adopt the code provided by Pytorch (Paszke et al., 2019)[6].

### A.1  NETWORK ARCHITECTURE

For Vision Transformers, we use the default network architecture adopted in Touvron et al. (2021). We implemented Pre-LN BERT in Huggingface's Transformers package (Wolf et al., 2019) such that:

- Within the residual branch of each Transformer block, we positioned LayerNorm to precede both the multi-head attention (MHA) and multi-layer perception (MLP) modules.
- For the MLM classification head, we use only one fully-connected layer (shared with the embedding).

We implemented Post-LN BERT in Huggingface's Transformers package (Wolf et al., 2019) such that:

- For the MLM classification head, we use only one fully-connected layer (shared with the embedding).

### A.2  DETAILED TRAINING CONFIGURATIONS

**Vision Transformers.** We train Vision Transformers on the ImageNet-1k (Deng et al., 2009) dataset. When training Vision Transformers from scratch, we apply a maximum learning rate of $1 \times 10^{-3}$ and run the training for 300 epochs with a batch size of 1024. We use AdamW (Loshchilov & Hutter, 2017) as the optimizer. We use a cosine learning rate scheduler that decays to a minimum learning rate of $10^{-5}$ with 5 warm-up epochs.

**BERT pre-training.** We train BERT (Devlin et al., 2019; Xiong et al., 2020) on masked language modeling task. The model is trained on the English Wiki corpus as per the methods in Tan & Bansal (2020) for 220k iterations with 5k warmup steps and a batch size of 256. We use AdamW as the optimizer. We use a maximum learning rate of $2 \times 10^{-4}$ and a cosine learning rate scheduler which decreases the learning rate to $2 \times 10^{-5}$. Following Liu et al. (2019), we remove the next sentence prediction task and use a fixed sequence length of 128 for model pre-training.

**BERT fine-tuning.** For fine-tuning task of BERT on the GLUE (Wang et al., 2018) dataset, we train 3 epochs with a learning rate of $1 \times 10^{-4}$ and a batch size of 32 for all tasks. We report correlation for

---

[4]`https://github.com/facebookresearch/deit/tree/main`
[5]`https://github.com/airsplay/vokenization`
[6]`https://github.com/pytorch/vision/tree/main/references/classification`

the STS-B dataset and Matthews correlation coefficient for the CoLA dataset. Accuracy is reported for the remaining datasets.

**Convolutional neural networks.** We train ResNets (He et al., 2016) and WideResNets (Zagoruyko & Komodakis, 2017) on the ImageNet-1k dataset for 90 epochs using SGD with an initial learning rate of 0.1. We set the batch size to be 128. Learning rate is decreased by 10 times at epochs 30 and 60.

### A.3   DETAILS OF BASELINES

We provide our implementation details of knowledge inheritance (KI) (Qin et al., 2021) in this section. Given a training dataset denoted as $\mathcal{D} = (\mathbf{x}_i, \mathbf{y}_i)_{i=1}^n$, we define the total loss $\mathcal{L}_{\text{Total}}$ as:

$$\mathcal{L}_{\text{Total}}(f_L; f_S, \mathcal{D}) = \sum_{(\mathbf{x}_i, \mathbf{y}_i) \in \mathcal{D}} (1 - \alpha)\mathcal{L}_{\text{self}}(f_L(\mathbf{x}_i), \mathbf{y}_i) + \alpha \mathcal{L}_{\text{KI}}(f_L, f_S, \mathbf{x}_i)$$

where $\alpha$ is a scalar controlling the strength of KI; The functions $f_S$ and $f_L$ respectively represent the small source model and the large target model; The loss function $\mathcal{L}_{\text{self}}$ computes the standard training loss, such as cross-entropy, between the prediction $f_L(\mathbf{x}_i)$ and the actual label $\mathbf{y}_i$. For soft KI, we set $\mathcal{L}_{\text{KI}} = \text{KL}(f_L(\mathbf{x}_i) \| f_S(\mathbf{x}_i))$. For hard KI, we set $\mathcal{L}_{\text{KI}} = \text{KL}(f_L(\mathbf{x}_i) \| \mathbf{e}_{\arg \max f_S(\mathbf{x}_i)})$, where KL stands for Kullback–Leibler divergence, and $\mathbf{e}$ is the standard basis vector.

During the KI process, we start with an initial $\alpha$ value of 0.5 and linearly decrease it to zero.

## B   NOTATIONS AND BACKGROUNDS

In this section, we introduce basic notations in section B.1, the definition of some normalization layers in section B.2, lossless expansion in vector space in section B.3, lossless expansion for operators (layers) in section B.4, and the rule of consecutive application of lossless expansion methods for consecutive layers in section B.4.3.

### B.1   NOTATIONS

All vectors are assumed to be column vectors. We define $\mathbf{0}_d$ to be a zero vector of dimension $d$. We use bold-faced letters for vectors, matrices, and tensors. For a vector $\mathbf{v}$, let $\mathbf{v}[i]$ be its $i$-th entry and $\mathbf{v}[: i]$ be its first $i$ entries. For a matrix $\mathbf{M}$, let $\mathbf{M}[i, j]$, $\mathbf{M}[i, :]$, and $\mathbf{M}[:, j]$ be its $(i, j)$-th entry, $i$-th row, and $j$-th column, respectively. Moreover, let $\mathbf{M}[: i, :]$ and $\mathbf{M}[:, : j]$ be its first $i$ rows and first $j$ columns, respectively. We use $\mathbf{M}^\intercal$ to denote the matrix transpose of $\mathbf{M}$. We use $[n]$ where $n \in \mathbb{Z}_+$ to denote $\{1, \cdots, n\}$. We use $\texttt{Id}$ to denote identity mapping. We use $\texttt{Concat}[\cdot]$ to denote horizontal concatenation.

### B.2   MODEL LAYERS

In this section, we give the formal definition of LayerNorm $\text{LN}(\cdot)$ and RMS Norm $\text{RMS}(\cdot)$.

**Definition 1** (LayerNorm). *LayerNorm $LN(\cdot; \boldsymbol{\mu}, \boldsymbol{\beta}, \epsilon)$ of dimension $D$ is defined as:*

$$LN(\mathbf{x}; \boldsymbol{\mu}, \beta, \epsilon) = \frac{\mathbf{x} - \mathbb{E}[\mathbf{x}]}{\sqrt{\text{Var}[\mathbf{x}] + \epsilon}} \odot \boldsymbol{\mu} + \boldsymbol{\beta},$$

*where $\mathbf{x}, \boldsymbol{\mu}, \boldsymbol{\beta} \in \mathbb{R}^D$.*

**Definition 2** (RMSNorm). *RMS Norm $RMS(\cdot; \boldsymbol{\mu}, \epsilon)$ of dimension $D$ is defined as:*

$$RMS(\mathbf{x}; \boldsymbol{\mu}, \epsilon) = \frac{\mathbf{x}}{\sqrt{\frac{1}{D} \sum_{i=1}^{D} (\mathbf{x}[i])^2 + \epsilon}} \odot \boldsymbol{\mu},$$

*where $\mathbf{x}, \boldsymbol{\mu} \in \mathbb{R}^D$.*

**Remark.** *In neural networks, inputs of normalization layers are usually high dimension tensors. In this case, LayerNorm and RMSNorm normally apply to the last dimension separately.*

### B.3 LOSSLESS EXPANSION IN VECTOR SPACE

In this section, we first give the general definition of lossless expansion in vector space.

**Definition 3** (Lossless expansion in vector space). *Given $\mathcal{S}$ and $\mathcal{T}$ are two vector spaces where the dimensions satisfy $dim(\mathcal{T}) \geq dim(\mathcal{S})$, a vector space expansion $\mathcal{V} : \mathcal{S} \to \mathcal{T}$ is said to be lossless if it is invertible.*

**Remark.** *Note that the identity function* Id *is lossless with its inverse being itself.*

Then we give a few examples of lossless vector space expansions. These examples will also be used in LEMON.

**Example B.3.1** (Vector average expansion $\mathcal{V}_{\text{avg}}$). *Let $\mathbf{x} \in \mathbb{R}^{D_S}$ be a vector of dimension $D_S$ and its average $Avg(\mathbf{x}) = \mathbb{E}[\mathbf{x}] = \frac{1}{D_S} \sum_i^{D_S} \mathbf{x}[i]$. $\mathbf{x}^*_{avg}$ is called the average expanded $\mathbf{x}$ of dimension $D_T$ with $D_T \geq D_S$ if*

$$\mathbf{x}^*_{avg} = \mathcal{V}_{avg}(\mathbf{x}) = Concat \left[ \underbrace{\mathbf{x}^\mathsf{T}, \cdots, \mathbf{x}^\mathsf{T}}_{\lfloor D_T/D_S \rfloor}, \underbrace{Avg(\mathbf{x}), \cdots, Avg(\mathbf{x})}_{D_T \bmod D_S} \right]^\mathsf{T} \in \mathbb{R}^{D_T}.$$

**Example B.3.2** (Vector zero expansion $\mathcal{V}_{\text{zero}}$). *Let $\mathbf{x} \in \mathbb{R}^{D_S}$ be a vector of dimension $D_S$. $\mathbf{x}^*_{zero}$ is called the zero expanded $\mathbf{x}$ of dimension $D_T$ with $D_T \geq D_S$ if*

$$\mathbf{x}^*_{zero} = \mathcal{V}_{zero}(\mathbf{x}) = Concat \left[ \underbrace{\mathbf{x}^\mathsf{T}, \cdots, \mathbf{x}^\mathsf{T}}_{\lfloor D_T/D_S \rfloor}, \underbrace{0, \cdots, 0}_{D_T \bmod D_S} \right]^\mathsf{T} \in \mathbb{R}^{D_T}.$$

**Example B.3.3** (Vector circular expansion $\mathcal{V}_{\text{circ}}$). *Let $\mathbf{x} \in \mathbb{R}^{D_S}$ be a vector of dimension $D_S$. $\mathbf{x}^*_{circ}$ is called the circular expanded $\mathbf{x}$ of dimension $D_T$ with $D_T \geq D_S$ if*

$$\mathbf{x}^*_{circ} = \mathcal{V}_{circ}(\mathbf{x}) = Concat \left[ \underbrace{\mathbf{x}^\mathsf{T}, \cdots, \mathbf{x}^\mathsf{T}}_{\lfloor D_T/D_S \rfloor}, \mathbf{x}^\mathsf{T}[: D_T \bmod D_S] \right]^\mathsf{T} \in \mathbb{R}^{D_T}.$$

**Example B.3.4** (Vector random expansion $\mathcal{V}_{\text{rand}}$). *Let $\mathbf{x} \in \mathbb{R}^{D_S}$ be a vector of dimension $D_S$. $\mathbf{x}^*_{rand}$ is called the random expanded $\mathbf{x}$ of dimension $D_T$ with $D_T \geq D_S$ if*

$$\mathbf{x}^*_{rand} = \mathcal{V}_{rand}(\mathbf{x}; \boldsymbol{\zeta}) = Concat \left[ \underbrace{\mathbf{x}^\mathsf{T}, \cdots, \mathbf{x}^\mathsf{T}}_{\lfloor D_T/D_S \rfloor}, \boldsymbol{\zeta}^\mathsf{T} \right]^\mathsf{T} \in \mathbb{R}^{D_T},$$

*where $\boldsymbol{\zeta} \in \mathbb{R}^{D_T \bmod D_S}$ is an arbitrary vector.*

**Remark.** *(1) All vector expansion examples above follow the same pattern. Specifically, when expanding from dimension $D_S$ to $D_T$, all vector expansion methods pad first $\lfloor D_T/D_S \rfloor D_S$ entries by repeating $\mathbf{x} \lfloor D_T/D_S \rfloor$ number of times. Each method deals with the remaining $D_T \bmod D_S$ entries differently. (2) The random vector $\boldsymbol{\zeta}$ in vector random expansion is arbitrary, so $\mathcal{V}_{avg}, \mathcal{V}_{zero}, \mathcal{V}_{circ} \subset \mathcal{V}_{rand}$. (3) Here all three examples are expansion methods for vectors. In practice, neural networks like Transformers are dealing high dimensional tensors. These tensors can essentially be thought of as collections of vectors. In such scenarios, we can apply the expansion methods separately to the last dimension of these tensors.*

In the following claim, we show that vectors expanded by these operators are lossless.

**Claim 1.** *Vector average expansion $\mathcal{V}_{avg}$, vector zero expansion $\mathcal{V}_{zero}$, vector circular expansion $\mathcal{V}_{circ}$, and vector random expansion $\mathcal{V}_{rand}$ are all lossless expansion for vectors.*

*Proof.* The inverse function $\mathcal{V}^{-1} : \mathbb{R}^{D_T} \to \mathbb{R}^{D_S}$ of these vector expansion methods is
$$\mathcal{V}^{-1}(\mathbf{x}) = \mathbf{x}[: D_S].$$

$\square$

**Remark.** *In practice, we want inverse mapping of expansion methods to be easily computed just like the example above.*

## B.4 LOSSLESS EXPANSION FOR OPERATORS

We then give the definition of lossless expansion for operators. These operators apply on tensors, hence our definition of lossless operator expansion is based on lossless expansion in vector space. These operators can be different layers used in Transformer architectures, including LayerNorm, convolutional layers, and fully-connected layers, etc.

**Definition 4** (Lossless expansion for operators). *Consider vector spaces $\mathcal{S}^{in}, \mathcal{S}^{out}, \mathcal{T}^{in}$ and $\mathcal{T}^{out}$ such that $dim(\mathcal{S}^{in}) \leq dim(\mathcal{T}^{in})$ and $dim(\mathcal{S}^{out}) \leq dim(\mathcal{T}^{out})$. Moreover, suppose the operator is denoted with $g(\cdot) : \mathcal{S}^{in} \to \mathcal{S}^{out}$. We say the operator expansion $\mathcal{E}$ is $(\mathcal{V}_{in}, \mathcal{V}_{out})$-lossless for $g(\cdot)$ if there exist lossless input vector space expansion $\mathcal{V}_{in} : \mathcal{S}^{in} \to \mathcal{T}^{in}$ and lossless output vector space expansion $\mathcal{V}_{out} : \mathcal{S}^{out} \to \mathcal{T}^{out}$ such that $\mathcal{V}_{out}(g(\mathbf{x})) = \mathcal{E}[g](\mathcal{V}_{in}(\mathbf{x})), \forall \mathbf{x} \in \mathcal{S}^{in}$.*

**Remark.** *(1) Intuitively, a lossless operator expansion can be understood as follows: when using $\mathcal{V}_{in}$ losslessly expanded input, the output of the $\mathcal{E}$ expanded operator is also a $\mathcal{V}_{out}$ losslessly expanded version of the original output. (2) For conciseness, we use '$\mathcal{E}[g]$ is $(\mathcal{V}_{in}, \mathcal{V}_{out})$-lossless' and '$\mathcal{E}$ is $(\mathcal{V}_{in}, \mathcal{V}_{out})$-lossless for $g(\cdot)$' interchangeably. (3) We only require the vector expansions $\mathcal{V}_{in}$ and $\mathcal{V}_{out}$ to be invertible, we do not have restrictions on the operator expansion $\mathcal{E}$.*

### B.4.1 LOSSLESS EXPANSION FOR MATRIX MULTIPLICATION

Then we give a few examples of lossless expansion for operators. We give examples for matrix multiplication since fully-connected layers are building blocks for Transformers. We first start by introducing the following three lossless operator expansion methods for matrix multiplication assuming that the input dimension is unchanged so $\mathcal{V}_{in} = \texttt{Id}$.

**Example B.4.1** (Matrix row-average expansion $\mathcal{E}_{row,avg}$). *Let $\mathbf{M} \in \mathbb{R}^{D_S \times P}$ be a matrix of dimension $D_S \times P$ and its row average $\mathbf{m} = \frac{1}{D_S} \sum_i^{D_S} \mathbf{M}[i,:]$. $\mathbf{M}^*_{row,avg}$ is called the row-average expanded $\mathbf{M}$ of dimension $D_T \times P$ with $D_T \geq D_S$ if*

$$\mathbf{M}^*_{row,avg} = \mathcal{E}_{row,avg}(\mathbf{M}) = \texttt{Concat} \left[ \underbrace{\mathbf{M}^\mathsf{T}, \cdots, \mathbf{M}^\mathsf{T}}_{\lfloor D_T/D_S \rfloor}, \underbrace{\mathbf{m}, \cdots, \mathbf{m}}_{D_T \bmod D_S} \right]^\mathsf{T} \in \mathbb{R}^{D_T \times P}.$$

*Moreover, $\mathcal{E}_{row,avg}$ is $(\texttt{Id}, \mathcal{V}_{avg})$-lossless for $\mathbf{M}$.*

**Example B.4.2** (Matrix row-zero expansion $\mathcal{E}_{row,zero}$). *Let $\mathbf{M} \in \mathbb{R}^{D_S \times P}$ be a matrix of dimension $D_S \times P$. $\mathbf{M}^*_{row,zero}$ is called the row-zero expanded $\mathbf{M}$ of dimension $D_T \times P$ with $D_T \geq D_S$ if*

$$\mathbf{M}^*_{row,zero} = \mathcal{E}_{row,zero}(\mathbf{M}) = \texttt{Concat} \left[ \underbrace{\mathbf{M}^\mathsf{T}, \cdots, \mathbf{M}^\mathsf{T}}_{\lfloor D_T/D_S \rfloor}, \underbrace{\mathbf{0}_P, \cdots, \mathbf{0}_P}_{D_T \bmod D_S} \right]^\mathsf{T} \in \mathbb{R}^{D_T \times P}.$$

*Moreover, $\mathcal{E}_{row,zero}$ is $(\texttt{Id}, \mathcal{V}_{zero})$-lossless for $\mathbf{M}$.*

**Example B.4.3** (Matrix row-circular expansion $\mathcal{E}_{row,circ}$). *Let $\mathbf{M} \in \mathbb{R}^{D_S \times P}$ be a matrix of dimension $D_S \times P$. $\mathbf{M}^*_{row,circ}$ is called the row-circular expanded $\mathbf{M}$ of dimension $D_T \times P$ with $D_T \geq D_S$ if*

$$\mathbf{M}^*_{row,circ} = \mathcal{E}_{row,circ}(\mathbf{M}) = \texttt{Concat} \left[ \underbrace{\mathbf{M}^\mathsf{T}, \cdots, \mathbf{M}^\mathsf{T}}_{\lfloor D_T/D_S \rfloor}, (\mathbf{M}[: D_T \bmod D_S, :])^\mathsf{T} \right]^\mathsf{T} \in \mathbb{R}^{D_T \times P}.$$

*Moreover, $\mathcal{E}_{row,avg}$ is $(\texttt{Id}, \mathcal{V}_{circ})$-lossless for $\mathbf{M}$.*

**Remark.** *Similar to vector expansion examples, these matrix row-expansion methods follow the same pattern. Specifically, when expanding the number of rows from dimension $D_S$ to $D_T$, these expansion methods pad first $\lfloor D_T/D_S \rfloor D_S$ rows by repeating $\mathbf{M}$ $\lfloor D_T/D_S \rfloor$ number of times. Each method deals with the remaining $D_T \bmod D_S$ rows differently.*

The following two lossless operator expansion methods assume that the output dimension is unchanged so $\mathcal{V}_{out} = \texttt{Id}$.

**Example B.4.4** (Matrix column-random expansion $\mathcal{E}_{\text{col,rand}}$). *Let $\mathbf{M} \in \mathbb{R}^{P \times D_S}$ be a matrix of dimension $P \times D_S$ and $\boldsymbol{\zeta} \in \mathbb{R}^{P \times (D_T \bmod D_S)}$ is an arbitrary matrix. $\mathbf{M}^*_{col,rand}$ is called the column-random expanded $\mathbf{M}$ of dimension $P \times D_T$ with $D_T \geq D_S$ if*

$$\mathbf{M}^*_{col,rand} = \mathcal{E}_{col,rand}(\mathbf{M}; \boldsymbol{\zeta}) = \texttt{Concat}\left[\underbrace{\mathbf{M}^1, \cdots, \mathbf{M}^{\lfloor D_T/D_S \rfloor}}_{\lfloor D_T/D_S \rfloor}, \boldsymbol{\zeta}\right] \in \mathbb{R}^{P \times D_T},$$

*where*

$$\sum_i^{\lfloor D_T/D_S \rfloor} \mathbf{M}^i = \mathbf{M}.$$

*Moreover, $\mathcal{E}_{col,rand}$ is $(\mathcal{V}_{zero}, \texttt{Id})$-lossless for $\mathbf{M}$.*

**Example B.4.5** (Matrix column-circular expansion $\mathcal{E}_{\text{col,circ}}$). *Let $\mathbf{M} \in \mathbb{R}^{P \times D_S}$ be a matrix of dimension $P \times D_S$ and $\mathbf{M}^{res} = \mathbf{M}[:,: D_T \bmod D_S] \in \mathbb{R}^{P \times (D_T \bmod D_S)}$. $\mathbf{M}^*_{col,circ}$ is called the column-circular expanded $\mathbf{M}$ of dimension $P \times D_T$ with $D_T \geq D_S$ if*

$$\mathbf{M}^*_{col,circ} = \mathcal{E}_{col,circ}(\mathbf{M}) = \texttt{Concat}\left[\underbrace{\mathbf{M}^1, \cdots, \mathbf{M}^{\lfloor D_T/D_S \rfloor}}_{\lfloor D_T/D_S \rfloor}, \mathbf{M}^{res}\right] \in \mathbb{R}^{P \times D_T},$$

*where*

$$\mathbf{M}^{res} + \sum_{i=1}^{\lfloor D_T/D_S \rfloor} \mathbf{M}^i[:,: D_T \bmod D_S] = \mathbf{M}[:,: D_T \bmod D_S],$$

*and*

$$\sum_{i=1}^{\lfloor D_T/D_S \rfloor} \mathbf{M}^i[:, D_T \bmod D_S :] = \mathbf{M}[:, D_T \bmod D_S :].$$

*Moreover, $\mathcal{E}_{col,rand}$ is $(\mathcal{V}_{circ}, \texttt{Id})$-lossless for $\mathbf{M}$.*

Note that lossless matrix row expansion and lossless matrix column expansion can be used together with the following claim.

**Claim 2.** *Consider matrix column expansion $\mathcal{E}_{col}$ is $(\mathcal{V}_{col}, \texttt{Id})$-lossless for $\mathbf{M}$, and matrix row expansion $\mathcal{E}_{row}$ is $(\texttt{Id}, \mathcal{V}_{row})$-lossless for $\mathbf{M}$. $\mathcal{E}_{col} \circ \mathcal{E}_{row}$ and $\mathcal{E}_{row} \circ \mathcal{E}_{col}$ are both $(\mathcal{V}_{col}, \mathcal{V}_{row})$-lossless for $\mathbf{M}$.*

The claim is easy to prove since rows and columns are expanded independently.

### B.4.2 LOSSLESS EXPANSION FOR BIAS

Note that the fully-connected layer consists of a matrix multiplication followed by a bias operator. We now give examples for the bias operator $\mathcal{B}(\mathbf{x}; \mathbf{b}) = \mathbf{x} + \mathbf{b}$.

**Example B.4.6** (Bias average expansion $\mathcal{E}_{\text{bias,avg}}$). *Consider the bias operator $\mathcal{B}(\mathbf{x}; \mathbf{b}) = \mathbf{x} + \mathbf{b}$ where $\mathbf{b} \in \mathbb{R}^{D_S}$. $\mathcal{B}^*_{bias,avg}(\cdot; \mathbf{b}^*_{bias,avg}) = \mathcal{E}_{bias,avg}[\mathcal{B}(\cdot; \mathbf{b})]$ is called the average expanded $\mathcal{B}$ of dimension $D_T$ with $D_T \geq D_S$ if $\mathbf{b}^*_{bias,avg} = \mathcal{V}_{avg}(\mathbf{b})$. Moreover, $\mathcal{E}_{bias,avg}$ is $(\mathcal{V}_{avg}, \mathcal{V}_{avg})$-lossless for $\mathcal{B}$.*

**Remark.** *Note that we can easily extend $\mathcal{E}_{bias,avg}$ to $\mathcal{E}_{bias,circ}$ and $\mathcal{E}_{bias,zero}$ by expanding $\mathbf{b}$ to $\mathcal{V}_{circ}(\mathbf{b})$ and $\mathcal{V}_{zero}(\mathbf{b})$, respectively. Moreover, $\mathcal{E}_{bias,circ}$ and $\mathcal{E}_{bias,zero}$ are $(\mathcal{V}_{circ}, \mathcal{V}_{circ})$-lossless and $(\mathcal{V}_{zero}, \mathcal{V}_{zero})$-lossless for $\mathcal{B}$, respectively.*

### B.4.3 CONSECUTIVE APPLICATION OF LOSSLESS EXPANSION FOR OPERATORS

In previous sections we give examples of lossless expansion methods for single operators. Now, to ensure lossless when applying expansion methods to consecutive layers/operators, we introduce the following claim:

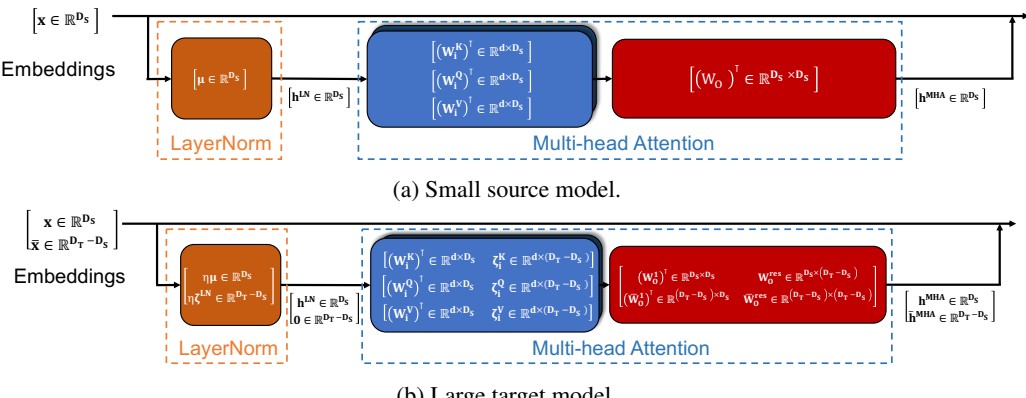

(a) Small source model.

(b) Large target model.

Figure 9: Illustration of LayerNorm expansion $\mathcal{E}_{\text{LN}}$ and MHA expansion $\mathcal{E}_{\text{MHA}}$. We assume $d = d_K = d_V$. We transpose weight matrices so that they can be considered left multiplied with vectors. The vectors in black font color indicate the intermediate values of inputs while the matrices in white color indicate parameters of the module. Biases are ignored for better illustration.

**Claim 3** (Lossless of consecutive application). *If $\mathcal{E}_1$ is $(\mathcal{V}_a, \mathcal{V}_b)$-lossless for $g_1$ and $\mathcal{E}_2$ is $(\mathcal{V}_b, \mathcal{V}_c)$-lossless for $g_2$. Then $\mathcal{E}_2[g_2] \circ \mathcal{E}_1[g_1]$ is $(\mathcal{V}_a, \mathcal{V}_c)$-lossless for $g_2 \circ g_1$.*

*Proof.* This is easily obtained if input $\mathbf{x}$ is $\mathcal{V}_a$ losslessly expanded, then the output of $\mathcal{E}_1[g_1](\cdot)$, $\mathbf{x}_{\text{mid}} = \mathcal{E}_1[g_1](\mathcal{V}_a(\mathbf{x}))$, is $\mathcal{V}_b$ lossless by definition. Using the fact that $\mathcal{E}_2[g_2](\cdot)$ is $(\mathcal{V}_b, \mathcal{V}_c)$-lossless and the input $\mathbf{x}_{\text{mid}}$ is $\mathcal{V}_b$ losslessly expanded, we conclude the proof. □

**Remark.** *By leveraging Claim 3, we can separately apply lossless expansion methods to various layers/operators in a larger network. The only requirement is that the output vector space expansion of one expansion method matches the input vector space expansion of the subsequent expansion method.*

## C DETAILS OF LEMON FOR PRE-LN TRANSFORMERS

In this section, we provide detailed explanation of applying LEMON on Pre-LN Transformer architecture. By Claim 3, we can deal with different modules separately. In the following sections, we delve into the details of applying expansion methods to these modules.

### C.1 WIDTH EXPANSION FOR PRE-LN TRANSFORMER BLOCKS

We first recap the Pre-LN Transformer architecture. It usually consists of (1) the embedding layer, (2) several Pre-LN Transformer blocks, (3) the final LayerNorm layer, and (4) a decoder layer.

Suppose that the hidden dimension $D$ of the transformer is increased from $D_S$ to $D_T$. The head dimension $d$ is unchanged during expansion. Hence, the number of heads is increased from $D_S/d$ to $D_T/d$. We use $\mathbf{W}_i^K, \mathbf{W}_i^Q, \mathbf{W}_i^V$ to denote the key, query, and value weight matrix for $i$-th head Head$_i$ in the MHA module. We use $W_O$ to denote the projection matrix.

We use $\mathcal{E}_{\text{block}}$ to denote the width expansion of Pre-LN Transformer blocks. $\mathcal{E}_{\text{block}}$ can be decomposed into (1) LayerNorm expansion $\mathcal{E}_{\text{LN}}$, (2) MHA module expansion $\mathcal{E}_{\text{MHA}}$, and (3) MLP module expansion $\mathcal{E}_{\text{MLP}}$. We introduce these expansion methods in the following paragraphs. We provide an illustration of $\mathcal{E}_{\text{LN}}$ and $\mathcal{E}_{\text{MHA}}$ in Figure 9.

**(1) LayerNorm expansion with $\mathcal{E}_{\text{LN}}$.** We define the expansion procedure for LN as follows. We use $\text{LN}(\cdot; \boldsymbol{\mu}_{\text{rand}}^*, \boldsymbol{\beta}_{\text{zero}}^*, \epsilon^*)$ where $\boldsymbol{\mu}_{\text{rand}}^* = \eta \mathcal{V}_{\text{rand}}(\boldsymbol{\mu}) \in \mathbb{R}^{D_T}$, $\boldsymbol{\beta}_{\text{zero}}^* = \mathcal{V}_{\text{zero}}(\boldsymbol{\beta}) \in \mathbb{R}^{D_T}$, and $\epsilon^* = \eta^2 \epsilon$ with $\eta = \lfloor D_T/D_S \rfloor * (D_S/D_T)$ to expand the original LayerNorm layer $\text{LN}(\cdot; \boldsymbol{\mu}, \boldsymbol{\beta}, \epsilon)$. The expansion is lossless and the proof is given in Proposition 1. Moreover, $\mathcal{E}_{\text{LN}}$ is $(\mathcal{V}_{\text{avg}}, \mathcal{V}_{\text{zero}})$-lossless for $\text{LN}(\cdot)$. In Figure 9, we omit $\epsilon$ and $\boldsymbol{\beta}$ for better illustration.

**(2) MHA expansion with $\mathcal{E}_{\textbf{MHA}}$.** We explain how to expand MHA as follow:

- **$\mathbf{W}_i^K, \mathbf{W}_i^Q, \mathbf{W}_i^V$ in self attention.** We consider the affine transformations applied to a single token $\mathbf{x} \in \mathbb{R}^{D_S}$ [7] in a sequence in self attention in the form of $\mathbf{k}_i(\mathbf{x}; \mathbf{W}_i^K, \mathbf{b}_i^K) = (\mathbf{W}_i^K)^\intercal \mathbf{x} + \mathbf{b}_i^K$, $\mathbf{q}_i(\mathbf{x}; \mathbf{W}_i^Q, \mathbf{b}_i^Q) = (\mathbf{W}_i^Q)^\intercal \mathbf{x} + \mathbf{b}_i^Q$, and $\mathbf{v}_i(\mathbf{x}; \mathbf{W}_i^V, \mathbf{b}_i^V) = (\mathbf{W}_i^V)^\intercal \mathbf{x} + \mathbf{b}_i^V$ where $(\mathbf{W}_i^K)^\intercal, (\mathbf{W}_i^Q)^\intercal, (\mathbf{W}_i^V)^\intercal \in \mathbb{R}^{d_K \times D_S}$ and $\mathbf{b}_i^K, \mathbf{b}_i^Q, \mathbf{b}_i^V \in \mathbb{R}^{d_K}$.

  During expansion, we increase the dimension of $(\mathbf{W}_i^K)^\intercal, (\mathbf{W}_i^Q)^\intercal, (\mathbf{W}_i^V)^\intercal$ from $\mathbb{R}^{d_K \times D_S}$ to $\mathbb{R}^{d_K \times D_T}$, and $\mathbf{b}_i^K, \mathbf{b}_i^Q, \mathbf{b}_i^V$ unchanged. Since the number of rows for $(\mathbf{W}_i^K)^\intercal, (\mathbf{W}_i^Q)^\intercal, (\mathbf{W}_i^V)^\intercal$ is unchanged, we only increase the number of columns by applying column-random expansion $\mathcal{E}_{\text{col,rand}}$ defined in Example B.4.4 to its transpose for each head, i.e., we use $\left\{ \mathcal{E}_{\text{col,rand}} \left[ (\mathbf{W}_i^K)^\intercal; \boldsymbol{\zeta}_i^K \right] \right\}^\intercal$, $\left\{ \mathcal{E}_{\text{col,rand}} \left[ (\mathbf{W}_i^Q)^\intercal; \boldsymbol{\zeta}_i^Q \right] \right\}^\intercal$, and $\left\{ \mathcal{E}_{\text{col,rand}} \left[ (\mathbf{W}_i^V)^\intercal; \boldsymbol{\zeta}_i^V \right] \right\}^\intercal$ for the expanded weights of $\mathbf{W}_i^K, \mathbf{W}_i^Q$ and $\mathbf{W}_i^V$, where $\boldsymbol{\zeta}_i^K, \boldsymbol{\zeta}_i^Q, \boldsymbol{\zeta}_i^V \in \mathbb{R}^{d_k \times (D_T \bmod D_S)}$ are random matrices. Biases are unchanged.

- **Heads in self attention.** We increase the number of heads in a circular pattern. See Figure 3b for an illustration. Note that (1) When $\lfloor D_T/D_S \rfloor > 1$, we can set $\mathbf{W}^1, \cdots, \mathbf{W}^{\lfloor D_T/D_S \rfloor}$ differently for replicated heads to break weight symmetry; (2) Additionally, when $D_T \bmod D_S \neq 0$, random matrices $\boldsymbol{\zeta}_i^K, \boldsymbol{\zeta}_i^Q, \boldsymbol{\zeta}_i^V$ can be chosen differently for replicated heads to break weight symmetry. Please see Example B.4.4 for definitions of $\mathbf{W}^1, \cdots, \mathbf{W}^{\lfloor D_T/D_S \rfloor}$ and $\boldsymbol{\zeta}_i^K, \boldsymbol{\zeta}_i^Q, \boldsymbol{\zeta}_i^V$.

- **Projection matrix in self attention.** For the projection transformation in the form of $\mathbf{W}_O^\intercal \mathbf{x} + \mathbf{b}_O$ where $\mathbf{W}_O^\intercal \in \mathbb{R}^{D_S \times D_S}$ and $\mathbf{b}_O \in \mathbb{R}^{D_S}$, we use $\mathcal{E}_{\text{col,circ}}$ and $\mathcal{E}_{\text{row,avg}}$ defined in Example B.4.5 and Example B.4.1 to expand the weights and biases. Specifically, we use $\left\{ \mathcal{E}_{\text{col,circ}} \left[ \mathcal{E}_{\text{row,avg}} (\mathbf{W}_O^\intercal) \right] \right\}^\intercal \in \mathbb{R}^{D_T \times D_T}$ for the expanded weight of $\mathbf{W}_O$. We then use $\mathcal{V}_{\text{avg}}(\mathbf{b}_O) \in \mathbb{R}^{D_T}$ for the expanded bias of $\mathbf{b}_O$.

Moreover, $\mathcal{E}_{\text{MHA}}$ is $(\mathcal{V}_{\text{zero}}, \mathcal{V}_{\text{avg}})$-lossless for MHA$(\cdot)$.

**(3) MLP expansion with $\mathcal{E}_{\textbf{MLP}}$.** Consider the MLP in the form of $\text{MLP}(\mathbf{x}) = \mathbf{W}_{\text{fc2}} \sigma(\mathbf{W}_{\text{fc1}} \mathbf{x} + \mathbf{b}_{\text{fc1}}) + \mathbf{b}_{\text{fc2}}$ where $\sigma$ is the non-linear activation. We explain how to expand MLP as follow:

- For the first fully-connected layer, we increase the columns by random expansion and increase the rows by circular expansion. Specifically, we use $\mathcal{E}_{\text{col,rand}} \left[ \mathcal{E}_{\text{row,circ}} (\mathbf{W}_{\text{fc1}}) \right]$ and $\mathcal{V}_{\text{circ}}(\mathbf{b}_{\text{fc1}})$ for the expanded weight and bias.

- For the second fully-connected layer, we increase the columns by circular expansion and increase the rows by average expansion. Specifically, we use $\mathcal{E}_{\text{col,circ}} \left[ \mathcal{E}_{\text{row,avg}} (\mathbf{W}_{\text{fc2}}) \right]$ and $\mathcal{V}_{\text{avg}}(\mathbf{b}_{\text{fc2}})$ for the expanded weight and bias.

Moreover, $\mathcal{E}_{\text{MLP}}$ is $(\mathcal{V}_{\text{zero}}, \mathcal{V}_{\text{avg}})$-lossless for MLP$(\cdot)$.

## C.2 Width Expansion of Other Layers

In this section, we explain how to expand the rest of the layers, i.e., embedding layers and decoder layers.

**Embeddings expansion with $\mathcal{V}_{\textbf{avg}}$.** We first average expand the embedding for each token $\mathbf{x}$ by adding its average, i.e., with $\mathcal{V}_{\text{avg}}$. For Vision Transformers, we do so by adding averaged channels for patch embeddings.

**Decoder layer expansion with $\mathcal{E}_{\textbf{dec}}$.** For Vision Transformers, the decoder layer is a fully-connected layer with the form $\text{Dec}(\mathbf{x}) = \mathbf{W}_{\text{dec}} \mathbf{x} + \mathbf{b}$. We increase the rows of the matrix by applying column-random expansion to its transpose, i.e., we use $\mathcal{E}_{\text{col,rand}}(\mathbf{W}_{\text{dec}})$ for the expanded weights. The bias is unchanged.

---

[7] In the formulation of MHA in section 3, $\mathbf{W}_i^K, \mathbf{W}_i^Q, \mathbf{W}_i^V$ are right matrix multiplied with the input sequence matrix $\mathbf{X} \in \mathbb{R}^{E \times D_S}$. Here we use the form of $\mathbf{W}_i \mathbf{x}$ for better illustration.

For language models, the decoder layer is shared with the embedding layer. So we have to instead scale the weight and bias of the LayerNorm before the decoder layer by $1/\lfloor D_T/D_S \rfloor$. Moreover, $\mathcal{E}_{\text{dec}}$ is $(\mathcal{V}_{\text{zero}}, \texttt{Id})$-lossless for $\texttt{Dec}$.

## C.3 DEPTH EXPANSION

Depth expansion is explained in the section 4.

## C.4 PARAMETER CHOICES

We consider the case $D_T \le 2D_S$ for better illustration.[8] There are mainly the following parameters to choose for LEMON. For the non-divisible case, we set the random parameter $\zeta$ in the LayerNorm such that $\zeta \sim \text{Unif}(-1, 1)$. When using matrix column-random expansion $\mathcal{E}_{\text{C, rand}}$ for the indivisible case, we use $\zeta_{i,j} \overset{\text{iid}}{\sim} \mathcal{N}(0, 0.02^2)$.

**Vision transformers.** For the width expansion parameters of the Vision Transformers, we set $\mathbf{W}^{\text{res}}$ for indivisible case and $\mathbf{W}^2$ for divisible case to be $\frac{1}{2}\mathbf{W}_O^{\mathsf{T}} + \Phi$, where $\Phi \in \mathbb{R}^{D_S \times (D_T - D_S)}$ is randomly initialized and $\Phi_{i,j} \overset{\text{iid}}{\sim} \mathcal{N}(0, 0.02^2)$.

For the depth expansion parameters, we set the free parameters that are used to cancel out replicated neurons following the distribution $\mathcal{N}(0, 0.02^2)$.

**ResNets.** For the width expansion parameters of the ResNet, we set $\mathbf{W}^{\text{res}}$ for indivisible case and $\mathbf{W}^2$ for divisible case to be $\frac{1}{2}\mathbf{W}_O^{\mathsf{T}} + \Phi$, where $\Phi \in \mathbb{R}^{D_S \times (D_T - D_S)}$ is randomly initialized and $\Phi$ follow the distribution used by the default implementation.

**Language models.** For the width expansion parameters of BERT, we set $\mathbf{W}^{\text{res}}$ for indivisible case and $\mathbf{W}^2$ for divisible case to $\Phi$, where $\Phi \in \mathbb{R}^{D_S \times (D_T - D_S)}$ is randomly initialized and $\Phi_{i,j} \overset{\text{iid}}{\sim} \mathcal{N}(0, 0.002^2)$.

For the depth expansion parameters, we set the projection matrix of the MHA block and the second fully-connected layer of the MLP block to be zero matrices. Moreover, inspired by advanced knowledge initialization (AKI) (Chen et al., 2021a), we append heads/neurons from the next adjacent layer.[9]

# D LEMON FOR OTHER ARCHITECTURES

Though we haven't included experiments for Res-Post-Norm and Post-LN blocks in our main experiments, we show that LEMON is able to perform lossless model expansion for these scenarios. We then briefly discuss how to handle RMS norm (Zhang & Sennrich, 2019), which is used in LLaMa (Touvron et al., 2023). We also discuss how to apply LEMON on convolutional neural networks.

## D.1 RES-POST-NORM TRANSFORMERS

We consider the Transformer with the following architecture: (1) an embedding layer, (2) several Res-Post-Norm blocks, and (3) the final decoder layer.[10]

### D.1.1 WIDTH EXPANSION

The only difference between the expansion methods of Res-Post-Norm Transformers and Pre-LN Transformers is that we zero expand embedding vector for each token with $\mathcal{V}_{\text{zero}}$.

For the MHA and MLP modules, we use the exact same expansion introduced in section C.1, where it is $(\mathcal{V}_{\text{zero}}, \mathcal{V}_{\text{avg}})$-lossless for $\texttt{MHA}$ and $\texttt{MLP}$. Consequently, our expansion is $(\mathcal{V}_{\text{zero}}, \mathcal{V}_{\text{zero}})$-lossless

---

[8]In fact we only need to deal with such cases in our experiments.

[9]This is still lossless since the last layer is a left-multiplied with a zero matrix followed by adding a zero vector.

[10]We assume there is no final LayerNorm before the final decoder layer.

for Res-Post-Norm Transformer blocks. Since the last decoder expansion is $(\mathcal{V}_{\text{zero}}, \texttt{Id})$-lossless for `Dec`, our expansion method is strict lossless.

### D.1.2 DEPTH EXPANSION

For increasing depth, we only need to set the weights and bias of the LayerNorm for each added layer to be all zeros.

### D.2 POST-LN TRANSFORMERS

For Post-LN Transformers, we can only deal with divisible cases, i.e., $D_T \bmod D_S = 0$. Suppose $D_T/D_S = n$, in this case, all the embedding and outputs of modules (MLP and MHA) are duplicated $n$ times and hence lossless. The only difficulty is to deal with depth expansion.

**Depth expansion.** Suppose we are given a pre-trained Post-LN Transformer block $g_1(\mathbf{x}) = \texttt{LN}_1(\texttt{Module}_1(\mathbf{x}) + \mathbf{x}) = \boldsymbol{\mu}_1 \odot \texttt{Norm}(\texttt{Module}_1(\mathbf{x}) + \mathbf{x}) + \mathbf{b}_1$. First we expand $\texttt{Module}_1$ to $\texttt{Module}_1^{0,*}$ so that it outputs zeros. Then we can create two expanded layers $g_1^*, g_2^*$ where $g_1^*(\mathbf{x}^*) = \mathbf{1} \odot \texttt{Norm}(\texttt{Module}_1^{0,*}(\mathbf{x}^*) + \mathbf{x}^*) + \mathbf{0} = \texttt{Norm}(\mathbf{x}^*)$ and $g_2^*(\mathbf{x}^*) = \boldsymbol{\mu}_1^* \odot \texttt{Norm}(\texttt{Module}_1^*(\mathbf{x}^*) + \mathbf{x}^*) + \mathbf{b}_1^*$. It is easy to show that $g_2^* \circ g_1^*$ is lossless where we use the fact that $\texttt{Norm}(\texttt{Norm}(\mathbf{x})) = \texttt{Norm}(\mathbf{x})$.

### D.3 TRANSFORMERS WITH RMS NORM

RMS Norm (Zhang & Sennrich, 2019) is used by foundation models like LLaMa (Touvron et al., 2023) and Baichuan (Yang et al., 2023). See Definition 2 for the definition of RMS Norm. Suppose we want to expand the RMS Norm from dimension $D_S$ to $D_T$, we use the following expansion.

**RMS Norm expansion with $\mathcal{E}_{\textbf{RMS}}$.** We use $\texttt{RMS}(\cdot; \boldsymbol{\mu}_{\text{rand}}^*, \epsilon^*)$ where $\boldsymbol{\mu}_{\text{rand}}^* = \eta \mathcal{V}_{\text{rand}}(\boldsymbol{\mu}) \in \mathbb{R}^{D_T}$, and $\epsilon^* = \eta^2 \epsilon$ with $\eta = \lfloor D_T/D_S \rfloor * (D_S/D_T)$ to expand the original RMS Norm layer $\texttt{LN}(\cdot; \boldsymbol{\mu}, \boldsymbol{\beta}, \epsilon)$. The expansion is $(\mathcal{V}_{\text{zero}}, \mathcal{V}_{\text{zero}})$-lossless for $\texttt{RMS}(\cdot)$. The proof is provided in Proposition 4.

### D.4 CONVOLUTIONAL NEURAL NETWORKS: RESNET

We use $\texttt{Conv}(k \times k, C_{\text{in}}, C_{\text{out}})$ to denote convolutional layer with $C_{\text{in}}$ in-channels, $C_{\text{out}}$ out-channels, and kernel size $k \times k$. We assume the kernel weight is $\mathbf{W} \in \mathbb{R}^{C_{\text{out}} \times C_{\text{in}} \times k \times k}$ and bias $\mathbf{b} \in \mathbb{R}^{C_{\text{out}}}$. We use `BN` and `ReLU` to denote BatchNorm and ReLU, respectively. ResNet and WideResNet with more than 50 layers consist of multiple Bottleneck blocks, where there are 3 sub-blocks: (1) $\texttt{Conv}(D, D_S, 1 \times 1)$-`BN`-`ReLU`, (2) $\texttt{Conv}(D_S, D_S, 3 \times 3)$-`BN`-`ReLU`, and (3) $\texttt{Conv}(D_S, D, 1 \times 1)$-`BN` in the residual branch.

We consider expanding ResNet to WideResNet.

**Width expansion.** To apply width expansion, we do the following:

(1) For the first sub-block, increase the number of out-channels of the first convolutional layer from $D_S$ to $D_T$. Specifically, the expanded weight satisfies $\mathbf{W}^*[i, :, :, :] = \mathbf{W}[i \bmod D_S, :, :, :], \forall i \in [D_T]$, and $\mathbf{b}^*[i] = \mathbf{b}[i \bmod D_S], \forall i \in [D_T]$. The output of the convolutional layer will be also in a circular pattern in the channel dimension. This also holds true after the application of BatchNorm and ReLU since the statistics of BatchNorm are computed within channels.

(2) For the second sub-block, increase the number of out-channels and in-channels of the first convolutional layer from $D_S$ to $D_T$. We apply the same operation to the out-channels dimension similar to (1). For the in-channel dimension, we need to make sure that the weights of replicated channels sum up to the original weight. Specifically, suppose that the replicated channels indices are denoted $\mathcal{C}_z = \{i | i \bmod D_S = z\}$. Then we need to set $\sum_{i \in \mathcal{C}_k} \mathbf{W}^*[i, :, :, :] = \mathbf{W}[k, :, :, :]$ for lossless expansion. Moreover, we need to make sure $\mathbf{W}^*[i, a, b, c] \neq \mathbf{W}^*[j, a, b, c], \forall i, j \in \mathcal{C}_z, a \in [C_{\text{in}}], b \in [k], c \in [k], z \in [C_{\text{out}}]$ for symmetry breaking.

(3) For the last sub-block, increase the number of in-channels of the first convolutional layer from $D_S$ to $D_T$ similar to (2).

**Depth expansion.** For depth expansion, we simply set the weight and bias of the last BatchNorm layers in the increased layers to be zeros.

# E  PROOFS

## E.1  PROOFS FOR TRANSFORMERS WITH LAYERNORM

In this section, we first show that three main components $\mathcal{E}_{\text{LN}}$, $\mathcal{E}_{\text{MHA}}$, and $\mathcal{E}_{\text{MLP}}$ are lossless. Then, we prove that LEMON defined in Appendix C is lossless.

We first start by showing that our LayerNorm expansion $\mathcal{E}_{\text{LN}}$ defined in section C.1 is lossless.

**Proposition 1** (Lossless expansion for LayerNorm $\mathcal{E}_{\text{LN}}$). *Consider* $LN(\cdot; \boldsymbol{\mu}, \boldsymbol{\beta}, \epsilon)$ *of dimension* $D_S$ *where* $\boldsymbol{\mu}, \boldsymbol{\beta} \in \mathbb{R}^{D_S}$. *Define average expanded of* $\mathbf{x} \in \mathbb{R}^{D_S}$ *of dimension* $D_T$ *to be* $\mathbf{x}_{avg}^* = \mathcal{V}_{avg}(\mathbf{x}) \in \mathbb{R}^{D_T}$, *where* $D_T \geq D_S$. *If* $\boldsymbol{\mu}_{rand}^* = \eta \mathcal{V}_{rand}(\boldsymbol{\mu}) \in \mathbb{R}^{D_T}$, $\boldsymbol{\beta}_{zero}^* = \mathcal{V}_{zero}(\boldsymbol{\beta}) \in \mathbb{R}^{D_T}$, *and* $\epsilon^* = \eta^2 \epsilon$, *where* $\eta = \sqrt{\lfloor D_T/D_S \rfloor * (D_S/D_T)}$, *then*

$$LN(\mathbf{x}_{avg}^*; \boldsymbol{\mu}_{rand}^*, \boldsymbol{\beta}_{zero}^*, \epsilon^*) = \mathcal{V}_{zero}(LN(\mathbf{x}; \boldsymbol{\mu}, \boldsymbol{\beta}, \epsilon)).$$

*Proof.* Since $\mathbb{E}[\mathbf{x}_{\text{avg}}^*] = \frac{1}{D_T} \sum_i \mathbf{x}_{\text{avg}}^*[i] = \frac{1}{D_T} \left( \lfloor D_T/D_S \rfloor \sum_i^{D_S} \mathbf{x}[i] + (D_T \bmod D_S)\mathbb{E}[x] \right) = \mathbb{E}[x]$ and $\text{Var}[\mathbf{x}_{\text{avg}}^*] = \frac{1}{D_T} \left( \lfloor D_T/D_S \rfloor D_S \text{Var}[\mathbf{x}] + (D_T \bmod D_S) * 0 \right) = \eta^2 \text{Var}[\mathbf{x}]$,

- For $1 \leq i \leq \lfloor D_T/D_S \rfloor D_S$:

$$
\begin{aligned}
\text{LN}(\mathbf{x}_{\text{avg}}^*; \boldsymbol{\mu}_{\text{rand}}^*, \boldsymbol{\beta}_{\text{zero}}^*, \epsilon^*)[i] &= \frac{\mathbf{x}_{\text{avg}}^*[i] - \mathbb{E}[\mathbf{x}_{\text{avg}}^*]}{\sqrt{\text{Var}[\mathbf{x}_{\text{avg}}^*] + \epsilon^*}} \odot \boldsymbol{\mu}_{\text{rand}}^*[i] + \boldsymbol{\beta}_{\text{zero}}^*[i] \\
&= \frac{\mathbf{x}[i \bmod D_S] - \mathbb{E}[\mathbf{x}]}{\eta\sqrt{\text{Var}[\mathbf{x}] + \epsilon}} \odot \eta\boldsymbol{\mu}[i \bmod D_S] + \boldsymbol{\beta}[i \bmod D_S] \\
&= \mathcal{V}_{\text{zero}}(\text{LN}(\mathbf{x}; \boldsymbol{\mu}, \boldsymbol{\beta}, \epsilon))[i]
\end{aligned}
$$

- For $\lfloor D_T/D_S \rfloor D_S \leq i \leq D_T$:

$$
\begin{aligned}
\text{LN}(\mathbf{x}_{\text{avg}}^*; \boldsymbol{\mu}_{\text{rand}}^*, \boldsymbol{\beta}_{\text{zero}}^*, \epsilon^*)[i] &= \frac{\mathbf{x}_{\text{avg}}^*[i] - \mathbb{E}[\mathbf{x}_{\text{avg}}^*]}{\sqrt{\text{Var}[\mathbf{x}_{\text{avg}}^*] + \epsilon^*}} \odot \boldsymbol{\mu}_{\text{rand}}^*[i] + \boldsymbol{\beta}_{\text{zero}}^*[i] \\
&= \frac{\mathbb{E}[\mathbf{x}] - \mathbb{E}[\mathbf{x}]}{\eta\sqrt{\text{Var}[\mathbf{x}] + \epsilon}} \odot \eta\boldsymbol{\zeta}[i \bmod D_S] + 0 \\
&= 0 \\
&= \mathcal{V}_{\text{zero}}(\text{LN}(\mathbf{x}; \boldsymbol{\mu}, \boldsymbol{\beta}, \epsilon))[i].
\end{aligned}
$$

Hence, $\text{LN}(\mathbf{x}_{\text{avg}}^*; \boldsymbol{\mu}_{\text{rand}}^*, \boldsymbol{\beta}_{\text{zero}}^*, \epsilon^*) = \mathcal{V}_{\text{zero}}(\text{LN}(\mathbf{x}; \boldsymbol{\mu}, \boldsymbol{\beta}, \epsilon))$. $\qquad\square$

**Remark.** *When* $D_T$ *is divisible by* $D_S$, *then* $\eta = 1$. *Hence, it explains why simply circularly expanding LayerNorm is lossless in such a scenario.*

Proposition 1 naturally leads to the following corollary.

**Corollary 1.** $\mathcal{E}_{LN}$ *introduced in Definition 1 is* $(\mathcal{V}_{avg}, \mathcal{V}_{zero})$-*lossless for* $LN(\cdot)$.

Using Claim 3, we are ready to prove that $\mathcal{E}_{\text{MHA}}$ and $\mathcal{E}_{\text{MLP}}$ are lossless. We first show that $\mathcal{E}_{\text{MHA}}$ is lossless in Proposition 2.

**Proposition 2** (Lossless of $\mathcal{E}_{MHA}$). $\mathcal{E}_{MHA}$ *defined in section C.1 is* $(\mathcal{V}_{zero}, \mathcal{V}_{avg})$-*lossless for* MHA.

*Proof.* Consider a sequence input $\mathbf{X} \in \mathbb{R}^{E \times D_S}$ is expanded losslessly by $\mathcal{V}_{\text{zero}}$ to $\mathbf{X}_{\text{zero}}^* \in \mathbb{R}^{E \times D_T}$. We expand the source small MHA such that the target large model is $\text{MHA}^* = \mathcal{E}_{\text{MHA}}(\text{MHA})$.

We first check the key, query, and value of each head $\text{Head}_i^*$ such that $i \leq H = D_s/d$ for the large model $\text{MHA}^*$. We denote them as $\mathbf{K}_i^*, \mathbf{Q}_i^*, \mathbf{V}_i^* \in \mathbb{R}^{E \times d_K}$. Note that biases $\mathbf{b}_i^K, \mathbf{b}_i^Q, \mathbf{b}_i^V \in \mathbb{R}^{d_K}$ are not expanded. Hence, these outputs are identical to the output of

the small source model $\mathbf{K}_i, \mathbf{Q}_i, \mathbf{V}_i \in \mathbb{R}^{E \times d_K}$ since $(\mathbf{W}_i^K)^\intercal, (\mathbf{W}_i^Q)^\intercal, (\mathbf{W}_i^V)^\intercal$ are expanded by $\mathcal{E}_{C,\,\mathrm{rand}}$, which is $(\mathcal{V}_{\mathrm{zero}}, \mathtt{Id})$-lossless. Consequently, $\mathrm{Head}_i^* = \mathrm{Attention}(\mathbf{Q}_i^*, \mathbf{K}_i^*, \mathbf{V}_i^*) = \texttt{softmax}\left(\mathbf{Q}_i^* (\mathbf{K}_i^*)^\intercal / \sqrt{d_K}\right) \mathbf{V}_i^*$ is identical to the output of $i$-th head of the $\texttt{MHA}$ in the source small model, which is $\mathrm{Head}_i$.

Since heads are circularly expanded, the output of $\texttt{MHA}^*$ is also $\mathcal{V}_{\mathrm{circ}}$ lossless.

Finally, since $\mathbf{W}_O^\intercal$ is expanded by $\mathcal{E}_{\mathrm{col,circ}}$ and $\mathcal{E}_{\mathrm{row,avg}}$, which is $(\mathcal{V}_{\mathrm{circ}}, \mathcal{V}_{\mathrm{avg}})$-lossless. With the fact that bias $\mathbf{b}_O$ is not expanded (unchanged), we obtain the result that $\mathcal{E}_{\mathrm{MHA}}$ is $(\mathcal{V}_{\mathrm{zero}}, \mathcal{V}_{\mathrm{avg}})$-lossless for $\texttt{MHA}$. $\qquad\square$

We then show that $\mathcal{E}_{\mathrm{MLP}}$ is lossless in Proposition 3.

**Proposition 3** (Lossless of $\mathcal{E}_{\mathrm{MLP}}$). *This is easily obtained since the first fully-connected layer is $(\mathcal{V}_{zero}, \mathcal{V}_{circ})$-lossless. Hence, the output is $\mathcal{V}_{circ}$ losslessly expanded. After applying element-wise nonlinear activation, the output is still $\mathcal{V}_{circ}$ losslessly expanded. Since the second fully-connected layer is $(\mathcal{V}_{zero}, \mathcal{V}_{circ})$-lossless, we conclude the proof that $\mathcal{E}_{MLP}$ is $(\mathcal{V}_{zero}, \mathcal{V}_{avg})$-lossless for $\texttt{MLP}$.*

Hence, using Proposition 2 and Proposition 3 along with Claim 3, we obtain the following Corollary 2 and Corollary 3.

**Corollary 2.** *The expanded Pre-LN $\texttt{MHA}$ module $\mathcal{E}_{MHA}(\texttt{MHA}) \circ \mathcal{E}_{LN}(\texttt{LN})$ is $(\mathcal{V}_{avg}, \mathcal{V}_{avg})$-lossless for $\texttt{MHA} \circ \texttt{LN}$.*

*Proof.* Since $\mathcal{E}_{\mathrm{LN}}$ is $(\mathcal{V}_{\mathrm{avg}}, \mathcal{V}_{\mathrm{zero}})$-lossless for $\texttt{LN}$, and $\mathcal{E}_{\mathrm{MHA}}$ is $(\mathcal{V}_{\mathrm{zero}}, \mathcal{V}_{\mathrm{avg}})$-lossless for $\texttt{MHA}$. The result is obtained by Claim 3. $\qquad\square$

**Corollary 3.** *The expanded Pre-LN MLP module $\mathcal{E}_{MLP}(\texttt{MLP}) \circ \mathcal{E}_{LN}(\texttt{LN})$ is $(\mathcal{V}_{avg}, \mathcal{V}_{avg})$-lossless for $\texttt{MLP} \circ \texttt{LN}$.*

By incorporating the residual connection, we obtain the following corollary.

**Corollary 4.** *The expanded Pre-LN modules (Pre-LN $\texttt{MHA}$/$\texttt{MLP}$) with residual connections are $(\mathcal{V}_{avg}, \mathcal{V}_{avg})$-lossless for the original Pre-LN modules with residual connections.*

Once again using Claim 3, we naturally obtain the following corollary.

**Corollary 5.** *The width-expanded Pre-LN Transformer layer $\mathcal{E}_{block}$ is $(\mathcal{V}_{avg}, \mathcal{V}_{avg})$-lossless for g.*

Finally, by considering the embedding layers and encoder layers, we show that LEMON is lossless.

**Corollary 6.** *LEMON introduced in section C.1 is $(\mathtt{Id}, \mathtt{Id})$-lossless for Pre-LN Transformers, i.e., strict lossless or identical.*

*Proof.* Since embeddings are average expanded, the output of Pre-LN Transformer blocks are average expanded. Hence, outputs of the final $\texttt{LN}$ before the encoder is zero expanded. Since the decoder layer expansion is $(\mathcal{V}_{\mathrm{zero}}, \mathtt{Id})$-lossless for $\texttt{Dec}(\cdot)$, we obtain the result that LEMON is $(\mathtt{Id}, \mathtt{Id})$-lossless. $\qquad\square$

### E.2 Proofs for Transformers with RMS Norm

In this section, we show that $\mathcal{E}_{\mathrm{RMS}}$ defined in section D.3 is lossless.

**Proposition 4** (Lossless expansion for RMS Norm $\mathcal{E}_{\mathrm{RMS}}$). *Consider $RMS(\cdot; \boldsymbol{\mu}, \epsilon)$ of dimension $D_S$ where $\boldsymbol{\mu} \in \mathbb{R}^{D_S}$. Define zero expanded of $\mathbf{x} \in \mathbb{R}^{D_S}$ of dimension $D_T$ to be $\mathbf{x}_{zero}^* = \mathcal{V}_{zero}(\mathbf{x}) \in \mathbb{R}^{D_T}$, where $D_T \geq D_S$. If $\boldsymbol{\mu}_{rand}^* = \eta \mathcal{V}_{rand}(\boldsymbol{\mu}) \in \mathbb{R}^{D_T}$, and $\epsilon^* = \eta^2 \epsilon$, where $\eta = \sqrt{\lfloor D_T / D_S \rfloor * (D_S / D_T)}$, then*

$$RMS(\mathbf{x}_{zero}^*; \boldsymbol{\mu}_{rand}^*, \epsilon^*) = \mathcal{V}_{zero}(RMS(\mathbf{x}; \boldsymbol{\mu}, \epsilon)).$$

*Proof.*  • For $1 \le i \le \lfloor D_T/D_S \rfloor D_S$:

$$\text{RMS}(\mathbf{x}^*_{\text{zero}}; \boldsymbol{\mu}^*_{\text{rand}}, \epsilon^*)[i] = \frac{\mathbf{x}^*_{\text{zero}}[i]}{\sqrt{\frac{1}{D_T} \sum_{i=1}^{D_T} (\mathbf{x}^*_{\text{zero}})^2 + \epsilon^*}} \odot \boldsymbol{\mu}^*_{\text{rand}}[i]$$

$$= \frac{\mathbf{x}[i \bmod D_S]}{\sqrt{\frac{D_S \lfloor D_T/D_S \rfloor}{D_T} \frac{1}{D_S} \sum_{i=1}^{D_S} (\mathbf{x}[i])^2 + \eta^2 \epsilon}} \odot \eta \boldsymbol{\mu}[i \bmod D_S]$$

$$= \frac{\mathbf{x}[i \bmod D_S]}{\eta \sqrt{\frac{1}{D_S} \sum_{i=1}^{D_S} (\mathbf{x}[i])^2 + \epsilon}} \odot \eta \boldsymbol{\mu}[i \bmod D_S]$$

$$= \mathcal{V}_{\text{zero}}(\text{RMS}(\mathbf{x}; \boldsymbol{\mu}, \epsilon))[i].$$

• For $\lfloor D_T/D_S \rfloor D_S \le i \le D_T$:

$$\text{RMS}(\mathbf{x}^*_{\text{zero}}; \boldsymbol{\mu}^*_{\text{rand}}, \epsilon^*)[i] = \frac{\mathbf{x}^*_{\text{zero}}[i]}{\sqrt{\frac{1}{D_T} \sum_{i=1}^{D_T} (\mathbf{x}^*_{\text{zero}})^2 + \epsilon^*}} \odot \boldsymbol{\mu}^*_{\text{rand}}[i]$$

$$= \frac{0}{\sqrt{\frac{1}{D_T} \sum_{i=1}^{D_T} (\mathbf{x}^*_{\text{zero}})^2 + \epsilon^*}} \odot \boldsymbol{\mu}^*_{\text{rand}}[i]$$

$$= 0$$

$$= \mathcal{V}_{\text{zero}}(\text{RMS}(\mathbf{x}; \boldsymbol{\mu}, \epsilon))[i].$$

Hence, $\text{RMS}(\mathbf{x}^*_{\text{zero}}; \boldsymbol{\mu}^*_{\text{rand}}, \epsilon^*) = \mathcal{V}_{\text{zero}}(\text{RMS}(\mathbf{x}; \boldsymbol{\mu}, \epsilon))$. □

Proposition 4 naturally leads to the following corollary.

**Corollary 7.** *$\mathcal{E}_{RMS}$ introduced in section D.3 is $(\mathcal{V}_{zero}, \mathcal{V}_{zero})$-lossless for $RMS(\cdot)$.*

## F  ADDITIONAL EXPERIMENTS

### F.1  COMPARISON WITH LiGO

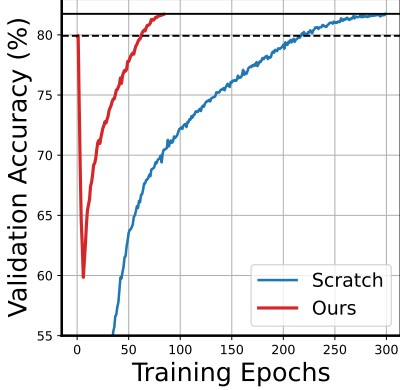

Figure 10: We expand ViT$(12, 384)$ to ViT$(12, 768)$. Our expanded model recovers the performance of the target model with 85 epochs (28.3% compared to training from scratch).

LiGO (Wang et al., 2023a) is unavailable for direct comparison due to the absence of open-source code. Hence, we compare them with their reported values. Note that our method is lossless only for Pre-LN Transformer architecture while LiGO reports their results for language models mainly on Post-LN BERT and RoBerTa. As a consequence, we compare our results with LiGO on ViT$(12, 384)$ (ViT-Small) $\rightarrow$ ViT$(12, 768)$ (ViT-Base).[11] The result is shown in Figure 10.

---

[11] Note that DeiT without distillation is exactly ViT.

Our method is able to recover the performance of the target model with 85 epochs, leading to a 71.67% computational saving. It is higher than the reported value of 55.40% for LiGO.[12]

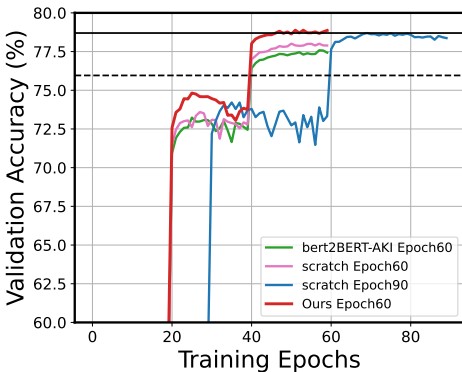

Figure 11: We expand ResNet-50 to WideResNet-110. Our expanded model (**Ours Epoch60; red**) recovers the performance of the target model within 60 epochs (33.3% compared to training from scratch). bert2BERT-AKI (**bert2BERT-AKI Epoch60; green**) is unable to accelerate the training compared training from scratch (**scratch Epoch60; pink**). **Note that LEMON is lossless. However, the accuracy of the model expanded by LEMON decreases after one epoch since there is no learning rate warm-up phase.**

We expand ResNet-50 to WideResNet-110 to assess the versatility and efficiency of LEMON in comparison to the bert2BERT-AKI method, known for its performance in the main manuscript. We utilized an optimized learning rate scheduler with a maximum rate of 0.1 (default), decaying at the 20th and 40th epochs.

**Results.** We show the result in Figure 11. LEMON is able to recover the performance of the large network in 60 epochs, achieving 33% computational savings. Note that bert2BERT-AKI shows inferior performance compared to training from scratch. We hypothesize that this might be due to a lack of compatibility of bert2BERT-AKI with the ResNet architecture

---

[12]Note that DeiT-Base (ViT-Base) has a final validation accuracy of 81.00% for LiGO, which is lower than the $\sim 81.70\%$ reported value of the official DeiT and our implementation.

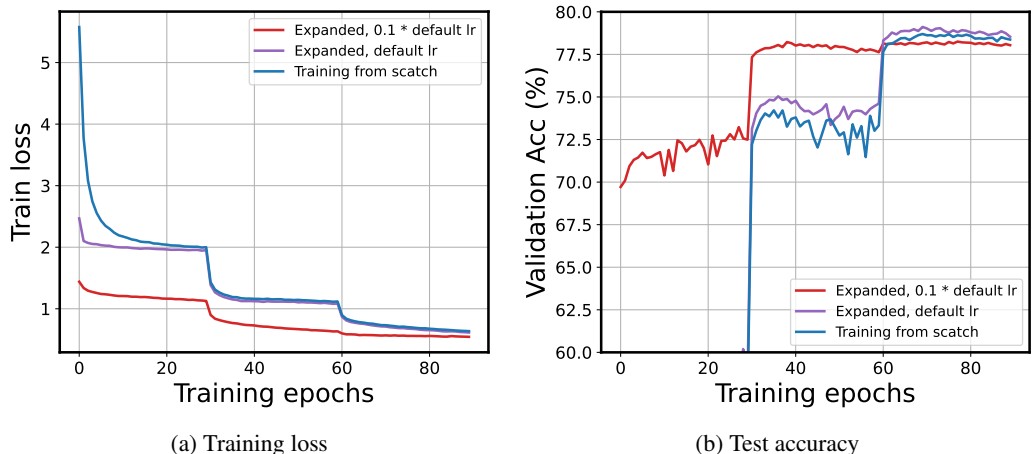

(a) Training loss            (b) Test accuracy

Figure 12: Training loss **(a)** and test accuracy **(b)** comparison of training from scratch with the maximum learning rate 0.1 (**Training from scratch; blue**), model expanded by LEMON trained with the maximum learning rate 0.1 (**Expanded, default lr; purple**), and model expanded by LEMON trained with the maximum learning rate 0.01 (**Expanded, 0.1 * default lr; red**). Using smaller learning rate leads to smaller training loss but worse generalization performance.

**Effects of maximum learning rate.** To understand how different maximum learning rates impact the performance of model expansion, we conducted similar experiments. Specifically, we compared the following setups: (1) Training a model from scratch with a maximum learning rate of 0.1, referred to as 'Training from scratch'; (2) A model expanded using LEMON and trained with the default maximum learning rate of 0.1, denoted as 'Expanded, default lr'; and (3) A model expanded using LEMON but trained with a reduced maximum learning rate of 0.01, termed 'Expanded, 0.1 * default lr'.

In line with the observations in Transformer architectures, we noticed that a smaller learning rate tends to result in lower training loss but potentially affects generalization performance adversely.

### F.3 POST-LN BERT

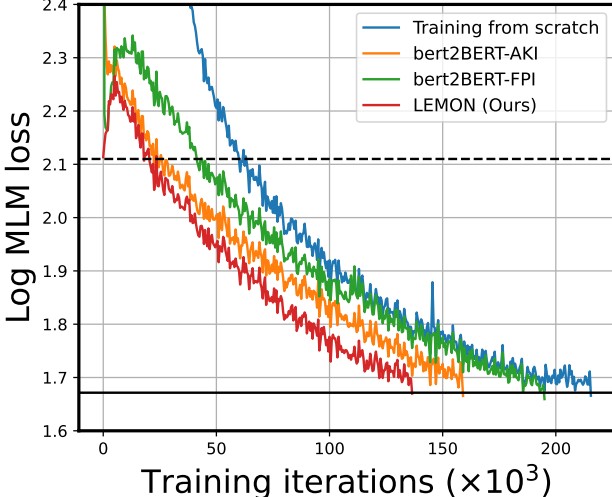

Figure 13: We expand Post-LN BERT$(6, 384)$ to BERT$(12, 768)$. Our expanded model achieves a log validation loss of 1.67 within 137k steps (63.43% compared to 216k steps for training from scratch).

In this section, we present our experiments conducted on Post-Layer Normalization (Post-LN) BERT models to further validate the effectiveness of LEMON. Specifically, we focused on expanding $BERT(6, 384)$ to $BERT(12, 768)$. We set a target log validation MLM loss of 1.67 for this experiment.

We trained the expanded model using LEMON for 143k steps. The results, as detailed in Figure 13, demonstrate that LEMON was able to achieve the targeted log validation MLM loss of 1.67 within just 137k steps. This result translates to a computational cost saving of 36.57%, compared to training $BERT(12, 768)$ from scratch.

## G  MORE RELATED WORKS

Efficiency in deep learning can be achieved in multiple ways. In this section we provide a brief overview of efficient deep learning regarding model training and inference, distinguishing it from methods addressing data efficiency (Gong et al., 2021; Wu et al., 2023a;b).

**Efficient deep learning.** In the realm of deep learning, the drive for efficiency has led researchers to develop a multitude of methods aimed at optimizing model efficiency. Techniques such as neural architecture search (NAS) (Zoph & Le, 2016; Liu et al., 2018) have been employed to automate the discovery of optimal network architecture. Quantization (Rastegari et al., 2016; Hubara et al., 2017) refines the numeric precision of model parameters to boost computational speed. Knowledge distillation (Hinton et al., 2015) and knowledge inheritance (Qin et al., 2021) allow target models to inherit the knowledge of their source counterparts. Neural network pruning (LeCun et al., 1989) involves removing unnecessary connections to accelerate model training or inference. Finally, model growth methods (Chen et al., 2015) directly use the weights of source models to initialize the large target models.

**Neural architecture search (NAS)** has emerged as a promising solution for automating the process of neural architecture design, eliminating the need for labor-intensive manual designs across various deep learning tasks. Initial methodologies leveraged reinforcement learning (Zoph & Le, 2016; Baker et al., 2016) and evolutionary algorithms (Real et al., 2019) to identify high-performing architectures. Despite their success, a significant drawback was their computational demands. Addressing this, DARTS (Liu et al., 2018) introduced a continuous relaxation of architectural representation, allowing for search via gradient descent. However, DARTS can be challenging to optimize, and its weight-sharing approach has been criticized for potential performance degradation (Yu et al., 2019; Wang et al., 2020b). Seeking further efficiency, Mellor et al. (Mellor et al., 2021) introduced a training-free NAS, which evaluates randomly initialized architectures, thus fully eliminating neural network training during the search phase. Subsequent training-free methods explored searches using Neural Tangent Kernel (NTK) (Xu et al., 2021; Chen et al., 2021b; Wang et al., 2022a), linear regions (Chen et al., 2021b), and criteria related to pruning (Abdelfattah et al., 2021).

When considered alongside model expansion, NAS holds potential for determining the optimal number of layers and hidden dimension of the large target model.

**Neural network pruning.** Pruning techniques can be broadly classified based on their timing into three categories: post-hoc pruning, pruning-at-initialization methods, and pruning-during-training methods. (1) Post-hoc pruning method removes certain weights of a fully-trained neural network. Post-hoc pruning was initially proposed to accelerate model inference (LeCun et al., 1989; Hassibi et al., 1993; Han et al., 2015), while lottery ticket works (Frankle & Carbin, 2018; Renda et al., 2020) shifted towards uncovering trainable sub-networks. (2) SNIP (Lee et al., 2018) is one of the pioneering works of pruning-at-initialization methods that aim to find trainable sub-networks without any training. Subsequent research (Wang et al., 2020a; Tanaka et al., 2020; de Jorge et al., 2020; Lee et al., 2019; Wang et al., 2022b) introduced varying metrics for pruning at the network initialization stage. (3) Finally, pruning-during-training methods prune or adjust DNNs throughout training. Early works incorporate explicit $\ell_0$ (Louizos et al., 2017) or $\ell_1$ (Wen et al., 2016) regularization terms to encourage sparsity, hence mitigating performance degradation commonly associated with post-hoc pruning. More recent techniques like DST methods (Bellec et al., 2017; Mocanu et al., 2018; Evci et al., 2020; Liu et al., 2021a; Wang et al., 2023b) allow for adaptive mask modifications during training while adhering to specified parameter constraints.

Neural network pruning has potential synergies with model expansion, akin to the dynamics of DST. A combined approach could involve iterative increases and decreases in hidden dimensions or layers during training, potentially accelerating training speed.

