# OpenReview forum: "LEMON: Lossless model expansion"
_ICLR.cc/2024/Conference — ICLR 2024 poster_

### Official Review · Reviewer_XnGd · 2023-10-31

**Soundness:** 3 good
**Presentation:** 4 excellent
**Contribution:** 3 good
**Rating:** 8
**Confidence:** 4

**Summary:**

This paper proposes a model expansion method that utilizes knowledge from existing smaller models. The authors analyze and design for different model structures, breaking the symmetry of repeating units by setting unequal output weights. This leads to a lossless model expansion approach. The training process is thoroughly analyzed, and the method achieves promising experimental results for both Vision and NLP Transformer models.

**Strengths:**

- The paper provides a comprehensive review of related work, presenting the current research status of expanding small models into larger ones.
- The method is highly versatile, as it is designed and analyzed for different structures within the Transformer, making it applicable to commonly used Transformer architectures. Moreover, they provide practical tuning suggestions for training.
- The analysis and observations made in the experiments are interesting.
- The authors demonstrate significant acceleration during the training process.

**Weaknesses:**

- After reading this paper, I would like to know more about the practical application of the model expansion method in real-world scenarios. Please provide more examples to illustrate the effectiveness and applicability of the proposed approach.
- How does the performance change in Figure 6b when using a learning rate larger than the default value?

**Questions:**

- Regarding the significant drop in accuracy during the early stages of training, it may be attributed to the transfer of pre-trained parameters from one local optimum to another. Considering the requirements of certain real-time systems, such performance drop in model accuracy is unacceptable.  I think maybe we can smooth out this process by, for example, setting a mask to control the number of trainable parameters in each epoch, gradually transitioning them to avoid a drastic drop in accuracy. Overall, solving this problem could be of significant importance for many real-world applications.

---

> ### Author Response · Authors · 2023-11-17
> **Response to reviewer XnGd**
>
> We are grateful for the reviewer’s recognition of our work's comprehensiveness, versatility, and significance.  Now we answer the reviewer's questions:
>
> > 1. After reading this paper, I would like to know more about the practical application of the model expansion method in real-world scenarios. Please provide more examples to illustrate the effectiveness and applicability of the proposed approach.
>
> We appreciate the reviewer's interest in the practical applications of our model expansion method. Our approach can be beneficial in various real-world scenarios:
>
> 1. **Streamlining model training across scales:** Companies often require models of different scales to meet diverse user needs or optimize inference speed. For instance, training models like Llama2 7B, 11B, and 70B typically requires substantial resources when done independently. Our method accelerates this process by allowing the sequential expansion of a smaller model (e.g., 7B) into larger versions (11B, then 70B). This approach significantly reduces computational resources and time.
>
> 2. **Dataset testing and adaptation:** When a new or modified dataset is introduced, it's common practice to test it with smaller-scale models before deploying larger ones. Our model expansion method enables an efficient transition from testing with a small model to full-scale deployment with a larger model.
>
> 3. **Efficient research prototyping:** Researchers often aim to build upon existing open-sourced models by scaling them up. Our method allows for the direct expansion of these pre-existing models into larger versions, facilitating prototyping and experimentation without the need for training large models from scratch.
>
> > 2. How does the performance change in Figure 6b when using a learning rate larger than the default value?
>
> We appreciate the reviewer’s question regarding the effect of using a maximum learning rate larger than the default value in Figure 6b. In our experiments, increasing the learning rate above the default value led to training instability and divergence. We believe that the default learning rate is often tuned to be close to the 'critical value' – a threshold beyond which the training becomes unstable.
>
> Indeed, this is an interesting topic, and we plan to delve deeper into this topic in our future work.
>
> > 3. Regarding the significant drop in accuracy during the early stages of training, it may be attributed to the transfer of pre-trained parameters from one local optimum to another. Considering the requirements of certain real-time systems, such performance drop in model accuracy is unacceptable. I think maybe we can smooth out this process by, for example, setting a mask to control the number of trainable parameters in each epoch, gradually transitioning them to avoid a drastic drop in accuracy. Overall, solving this problem could be of significant importance for many real-world applications.
>
> We thank the reviewer for raising an important concern regarding the initial accuracy drop in our method, particularly in the context of real-time systems where stable performance is crucial. Indeed, our method in its current form might not be ideally suited for such applications due to this challenge. And we are actively investigating solutions to this issue.
>
> Regarding the learning rate setting:
>
> (1) **Constant or minimally changing learning rate:** Our investigations indicate that the initial performance drop is primarily attributed to a sudden increase in the learning rate. In scenarios where the learning rate is constant or undergoes minimal changes, we have observed that this drop is significantly less pronounced. This indicates that our expansion method may be effectively utilized in such settings.
>
> (2) **Decreasing learning rate:** In cases where the learning rate decreases over time, we propose an interim solution of continuing to use the small/source model at the point of expansion. Once the expanded model gains its expected performance, the system can seamlessly transition to the expanded model, ensuring both stability and the benefits of expansion.
>
> Thank you for highlighting this critical aspect. It forms an important part of our ongoing research, and we are committed to developing more effective solutions to reduce the initial performance drop without compromising on training efficiency.

---

### Official Review · Reviewer_pAyq · 2023-10-31

**Soundness:** 3 good
**Presentation:** 3 good
**Contribution:** 3 good
**Rating:** 8
**Confidence:** 3

**Summary:**

This paper proposed a model expansion algorithm that uses the pre-trained parameters of a smaller model to initialize a larger model. The proposed algorithm allows expanding model’s width and depth to arbitrary width and depth for most Transformer variants. The algorithm ensures that the larger model has the same output as the smaller model (thus does not require calibration dataset) to preserve the small model performance while having symmetry breaking for continuous training to further optimize the larger model. The expansion algorithm alone does not ensure that the larger model can achieve the same performance as the same sized model trained from scratch using smaller training cost. The authors found right training configuration is critical for obtaining this goal.

**Strengths:**

1. The proposed method is simple, yet it allows expanding smaller model to arbitrary width and depth (not necessarily indivisible by the width and depth of the original models) while ensuring the output of the expanded model stays the same as smaller model and parameter symmetry breaking.

2. The expansion can be performed on individual modules of a Transformer. This localized expansion ensure compatibility with different Transformer variants.

3. The expansion algorithm alone does not give desired performance. The authors explored different training configuration, including learning rate and scheduler, for more desired performance and found the training configurations greatly affect the results.

4. The authors performed ablation study to isolate the effect of optimized training configuration from expansion algorithms to make sure the proposed expansion algorithm indeed performs better compared to baselines.

**Weaknesses:**

1. In Figure 7 (c) and (d), the loss curves for BERT language modeling are still decreasing when training is stopped. It might be better to train the model till convergence to evaluate whether or not the proposed method can have the same performance as the model trained from scratch.

2. Vision Transformer is a pre-norm Transformer, and in BERT language modeling, the authors also used the pre-norm variant. Since the authors claimed compatibility of the algorithm with different variants, it would be better to see the experiments on different variants (at least a post-norm variant) to verify the claim. While in Appendix, the authors show lossless expansion for other variants, it is also important to evaluate the performance metrics.

3. Since this work also studies the initialization of model parameters, it might be interesting to compare the proposed idea with other initialization approach, such as Mimetic initialization (https://arxiv.org/abs/2305.09828, also mentioned in the related work)

**Questions:**

The suggestions are listed in weakness section.

---

> ### Author Response · Authors · 2023-11-17
> **Response to reviewer pAyq**
>
> We are grateful for the reviewer’s appreciation of our work’s simplicity and effectiveness, as well as our study on training configurations. We now address the questions raised:
>
> > 1.  In Figure 7 \(c\) and (d), the loss curves for BERT language modeling are still decreasing when training is stopped. It might be better to train the model till convergence to evaluate whether or not the proposed method can have the same performance as the model trained from scratch.
>
> We appreciate the reviewer's observation concerning the training duration in these figures. The decision to stop training when the logarithmic validation MLM loss reaches 1.64 was intentional. This specific loss value was selected because, beyond this point, the benefits of continued training diminish due to the reduced learning rate. Additionally, the choice of a logarithmic scale for loss visualization can sometimes exaggerate the appearance of a decreasing trend, which is important to consider when interpreting these curves.
>
> This choice allows for a more stable comparison across methods, as the validation loss is subject to fluctuations due to the stochastic nature of MLM masks during training/validation.
>
> Regarding the comparison of LEMON's performance with the model trained from scratch, we refer to the results presented in Table 2. In this table, we compare the downstream performance of BERT(12,768) models expanded using LEMON from BERT(6,384)/BERT(6,512) over the **complete** 165k/132k training steps. The results demonstrate that the models expanded via LEMON outperform the performance of models trained from scratch for the **complete** 220k steps. This evidence strongly supports the effectiveness of LEMON.
>
> > 2. Vision Transformer is a pre-norm Transformer, and in BERT language modeling, the authors also used the pre-norm variant. Since the authors claimed compatibility of the algorithm with different variants, it would be better to see the experiments on different variants (at least a post-norm variant) to verify the claim. While in Appendix, the authors showlossless expansion for other variants, it is also important to evaluate the performance metrics.
>
> We are grateful for the reviewer's constructive feedback regarding the evaluation of LEMON with different Transformer variants. To address this point, we have expanded our experiments to include Post-LN BERT models, specifically testing the expansion from BERT(6,384) to BERT(12,768) as follow. We repoort number of training steps needed to achieve a log validation MLM loss of 1.67. (See details in Appendix F.3)
>
> | Method                | Training steps needed |
> |-----------------------|-----------------------|
> | Training from scratch |          216k         |
> |     bert2BERT-FPI     |          195k         |
> |     bert2BERT-AKI     |          159k         |
> |      LEMON (Ours)     |          **137k**         |
>
> The results demonstrate that LEMON not only outperforms these baselines but also achieves 36.57% less computational savings than training from scratch.
>
> > 3. Since this work also studies the initialization of model parameters, it might be interesting to compare the proposed idea with other initialization approach, such as Mimetic initialization (https://arxiv.org/abs/2305.09828), also mentioned in the related work.
>
> We appreciate the reviewer’s suggestion to compare LEMON with Mimetic initialization. We recognize that Mimetic initialization addresses a different scenario, primarily focusing on scenarios where no pre-trained (small) models are available. This distinction makes our work and Mimetic initialization **orthogonal yet potentially complementary**. For example, Mimetic initialization might be used to effectively train the initial small model, which could then be efficiently expanded using LEMON.
>
> At present, the lack of open-sourced code for Mimetic initialization limits our ability to conduct related experiments. However, we are keenly interested in this possibility and plan to explore such comparisons as soon as the code becomes available.

---

### Official Review · Reviewer_JSXE · 2023-10-31

**Soundness:** 3 good
**Presentation:** 3 good
**Contribution:** 3 good
**Rating:** 6
**Confidence:** 4

**Summary:**

**Idea**:
* This paper introduces a method for initializing scaled models using the weights of their smaller pre-trained counterparts. The method allows for expanding neural network models in a lossless manner, increasing depth and width without sacrificing performance.
* The paper introduces lossless layer expansion techniques, including row-average expansion, row-zero expansion, column-random expansion, and column-circular expansion.
* The expansion procedure for LayerNorm and Multi-head Attention (MHA) modules in Pre-LN Transformer blocks is explained, showing that the expansion is lossless and preserves the properties of the original layers.

**Experiments and Analysis**:
* The method is versatile and compatible with various network structures, although the _experiments are only shown on Vision Transformers and BERT_. LEMON outperforms baselines on these architectures in terms of performance and computational cost.
* Detailed explanations and insights into various techniques and approaches for training deep neural networks are provided, with a focus on language models.
* The authors investigate the effects of maximum learning rate and learning rate scheduler when training expanded models.
* LEMON is compared to a similar method called LiGO and shows better results in terms of computational saving.

**Strengths:**

### S1 - Interesting technical contributions
* The authors provide a comprehensive and detailed exploration of lossless model expansion techniques (e.g. row-average expansion, row-zero expansion, column-random expansion, and column-circular expansion), including addressing the challenges of symmetry breaking and indivisible width increments.
* Provide valuable insights into training recipes for expanded models, including an optimized learning rate scheduler that can further enhance performance.

### S2 - Good results and experimental analysis
* Extensive experiments with ViT and BERT are shown with a thorough investigation of the effects of maximum learning rate and learning rate scheduler when training expanded models.
* The proposed method achieves similar performance to the original models with fewer training epochs, highlighting its efficiency and effectiveness.
* LiGO is a similar very recent method, and LEMON shows better results in terms of computational saving.

**Weaknesses:**

### W1 - Experiments limited only to ViT and BERT
* The paper could benefit from experiments on the generalizability of LEMON to other architectures beyond Vision Transformers and BERT models. For example, CNN models are completely unexplored in terms of experiments.
* I suggest adding model expansion experiments for ResNet18 --> ResNet50 and EfficientNetB0 --> EfficientNetB4 (or other variants).

### W2 - Lacks theoretical analysis/explanation of "effect of learning rate and schedule"
* Sections 5.1 and 5.2 experimentally study the effect of learning rates and schedules. However, the paper lacks a theoretical analysis of why this happens. For example, why does a small learning rate lead to lower final performance? I think only experimental verification is not enough and this requires some theoretical analysis.

**Questions:**

Can "incremental" model expansion help achieve even better performance? For example, instead of expanding from "Model (small) pretrained --> Model (huge)", would it be better to expand in steps as "Model (small) pretrained --> Model (middle) --> Model (big) --> Model (huge)"

---

> ### Author Response · Authors · 2023-11-17
> **Response to reviewer JSXE**
>
> We thank the reviewers for acknowledging our method is versatile and interesting. We now address the reviewer's concerns:
>
> > 1. The paper could benefit from experiments on the generalizability of LEMON to other architectures beyond Vision Transformers and BERT models. For example, CNN models are completely unexplored in terms of experiments. I suggest adding model expansion experiments for ResNet18 --> ResNet50 and EfficientNetB0 --> EfficientNetB4 (or other variants).
>
> Thank your for your constructive feedback. In response to your suggestion, we have included experiments with expansion from ResNet-50 to WideResNet-110 in Appendix F.2. We compare LEMON with the best baseline method, i.e., bert2BERT-AKI. Our results indicate that LEMON is able to complete the training process within 60 epochs, which is 66.7% of the time required for training from scratch. Moreover, we observed that the bert2BERT-AKI baseline did not offer a similar acceleration, underscoring LEMON's advantages.
>
> Regarding the choice of models, we did not opt for the expansion from ResNet-18 to ResNet-50 because they utilize different architectural blocks (Basic block vs. BottleNeck block). Instead, we expand ResNet-50 to WideResNet-110 to showcase LEMON’s capabilities involving both depth and width expansion.
>
> > 2. Sections 5.1 and 5.2 experimentally study the effect of learning rates and schedules. However, the paper lacks a theoretical analysis of why this happens. For example, why does a small learning rate lead to lower final performance? I think only experimental verification is not enough and this requires some theoretical analysis.
>
> We thank the reviewer for highlighting the importance of theoretical analysis in understanding the effects of learning rates and schedules on model performance. Indeed, several theoretical works have attempted to shed light on the relationship between learning rate, training schedules, and model generalization. For instance, He et al. [1] establishes a generalization bound correlated with the batch size to learning rate ratio, and Li et al. [2] demonstrates that networks trained with a large initial learning rate and subsequent annealing can generalize better than networks trained with a consistently small learning rate, based on the learning order of the model.
>
> We believe that our study, primarily empirical in nature, can provide valuable, practical insights into this intricate relationship and can serve as a useful reference for theoreticians in the field.
>
> Recognizing the significance of a combined theoretical and empirical approach, we are committed to incorporating a deeper theoretical analysis into our future research, which may involve collaborations with theoreticians.
>
> > 3. Can "incremental" model expansion help achieve even better performance? For example, instead of expanding from "Model(small) pretrained --> Model (huge)", would it be better to expand in steps as "Model (small) pretrained --> Model (middle) -->Model (big) --> Model (huge)"
>
> We are grateful for the reviewer’s suggestion regarding the potential of 'incremental' model expansion. Indeed, this concept is an interesting aspect of model scaling and could potentially enhance performance. However, due to its complexity and the constraints of the rebuttal period, we have not yet explored this incremental approach. But we recognize the potential value of this method and are keen to investigate it in our future work.
>
> [1] He, Fengxiang, Tongliang Liu, and Dacheng Tao. "Control batch size and learning rate to generalize well: Theoretical and empirical evidence." Advances in neural information processing systems 32 (2019).
>
> [2] Li, Yuanzhi, Colin Wei, and Tengyu Ma. "Towards explaining the regularization effect of initial large learning rate in training neural networks." Advances in Neural Information Processing Systems 32 (2019).

---

### Official Review · Reviewer_u5r2 · 2023-11-01

**Soundness:** 3 good
**Presentation:** 3 good
**Contribution:** 2 fair
**Rating:** 6
**Confidence:** 4

**Summary:**

The authors proposed a lossless model expansion method which initialize scaled models using the weights of smaller pre-trained model. Specifically, the proposed method break the symmetry of replicated neurons by setting their fan-out weights to be unequal, and introduce average expansion to deal with LayerNorm for indivisible width increment. Besides, the authors explored the training recipes for the expanded models and proposed an optimized learning rate scheduler that decays more rapidly than training from scratch. Experimental results show that the proposed method can effectively expand both Vision Transformer and BERT, while significantly reducing the training overhead.

**Strengths:**

1. The motivation is clear. The author focus on scaling deep neural networks in effective way by leveraging the knowledge acquired by their smaller counterparts.
2. The paper is well organized in terms of written description. The authors provided easy-to-understand diagrams.
3. The idea is technically feasible and the authors provide detailed proofs in appendix.

**Weaknesses:**

1. The challenge arising with the ‘symmetry breaking’ is described in the third paragraph of section Introduction: “the expanded model will never gain more capacity than the source model.”This statement raises confusion as training a model with smaller capacity but larger size appears to be of limited value, which incurs greater overhead but achieves limited performance.
2. Have the considered baselines for expansion in Section 6 been confirmed to be lossless? If not, it is necessary to present the gap with the original model. If they are indeed lossless, an analysis should be provided to explain why the proposed method achieves higher validation accuracy compared to AKI, which also breaks symmetry, as shown in Figure 8.
3. The results in Table 2 are a bit confusing. It is unclear whether model expansion or longer training duration indeed contributes to the improved performance.
4. The novelty of this paper seems quite limited. The key idea of model expansion seems a simple extension of net2net. Are there any essential technical differences? In my opinion, a simple extension of an existing approach is insufficient for a top-tier conference.


After rebuttal:

Thanks for the detailed response from the authors. Although the key idea is similar to net2net, there are still some new contributions in expanding layernorm and attention. Thus, I decided to raise my score.

**Questions:**

Please refer to the weakness part.

---

> ### Author Response · Authors · 2023-11-17
> **Response to reviewer u5r2 [Part 1/2]**
>
> We thank the reviewer for acknowledging the clarity, organization, and feasibility of our work. We now address the concerns raised:
>
> > 1. The challenge arising with the ‘symmetry breaking’ is described in the third paragraph of section Introduction: “the expanded model will never gain more capacity than the source model.”This statement raises confusion as training a model with smaller capacity but larger size appears to be of limited value, which incurs greater overhead but achieves limited performance.
>
> We appreciate the reviewer's feedback and the opportunity to clarify the distinction between **'weight symmetry'** and **'break weight symmetry.'** We acknowledge that our initial explanation might have caused some confusion.
>
> In our paper, 'weight symmetry' refers to a situation where, after neuron replication, the weights associated with these replicated neurons are initially identical and continue to remain identical throughout the training process. Hence, 'weight symmetry' limits the neural network's representation power. To **avoid** 'weight symmetry,' '**breaking** weight symmetry' becomes crucial, where we set the fan-out weights of replicated neurons to be unequal.
>
> This will be more clearly articulated in the revised manuscript, particularly in the Section 1 and 3.  The terms 'weight symmetry' and 'break weight symmetry' are highlighted in blue.
>
> > 2. Have the considered baselines for expansion in Section 6 been confirmed to be lossless? If not, it is necessary to present the gap with the original model. If they are indeed lossless, an analysis should be provided to explain why the proposed method achieves higher validation accuracy compared to AKI, which also breaks symmetry, as shown in Figure 8.
>
> Thank you for your question. The baseline models, namely bert2BERT-AKI and bert2BERT-FPI, are **NOT** lossless. This is reflected in the MLM loss and accuracy for BERT and ViT, as shown below.
>
> | Model            | MLM Loss ($\downarrow$) of BERT(6,384)  | MLM Loss ($\downarrow$) of BERT(6,512) | Accuracy ($\uparrow$) of ViT(6,384) | Accuracy ($\uparrow$) of ViT(6,512)|
> | ---------------- | --------------- | -------------- | --------------- | -------------- |
> | Source model     | 8.20               | 6.97              | 73.82           | 76.61          |
> | bert2BERT-AKI    | 29217.87        | 902.83         | 15.91           | 33.48          |
> | bert2BERT-FPI    | 22.34           | 9.14           |  71.88           | 73.71          |
> | LEMON            | 8.14            | 6.95           | 73.82           | 76.61          |
>
> It's important to note that the MLM validation loss may slightly vary from the source model due to the different MLM masks applied in each batch. However, we have carefully verified LEMON's losslessness by comparing the outputs of small and large models across multiple data samples.
>
> To provide a clearer understanding of the effectiveness our model expansion approach, we have added data points in our plots to show performance right after expansion, before any additional training.
>
> > 3. The results in Table 2 are a bit confusing. It is unclear whether model expansion or longer training duration indeed contributes to the improved performance.
>
> We thank the reviewer for highlighting the need for clarity regarding the results in Table 2. Our aim is to compare the performance of **models expanded using LEMON (trained for 132/165k steps)**, with **models trained from scratch for a longer duration of 220k steps.** The improved performance is primarily due to the model expansion process, as the training steps for the expanded models are significantly fewer (132/165k) compared to the 220k steps for the models trained from scratch.
>
> We realize that our initial presentation might have led to some confusion. Specifically, the mention of 'A potential reason for this may be its longer training duration' refers to a **different comparison** – between LEMON expanded models using **BERT(6,384)** and **BERT(6,512)**.
>
> To avoid this confusion, we have revised Table 2 by including the total training steps for each case and modifying the caption.

---

> ### Author Response · Authors · 2023-11-17
> **Response to reviewer u5r2 [Part 2/2]**
>
> > 4. The novelty of this paper seems quite limited. The key idea of model expansion seems a simple extension of net2net. Are there any essential technical differences? In my opinion, a simple extension of an existing approach is insufficient for a top-tier conference.
>
> We thank the reviewer for their comments on the novelty of our paper. While we recognize the foundational role of Net2Net in the field of model expansion, we respectfully disagree that our work is a mere extension of it, particularly when considering Transformers with LayerNorm.
>
> **Width and depth expansion:**
> - Net2Net was initially developed for CNNs using ReLU activation and BatchNorm. While groundbreaking, it had its limitations. For instance, its depth expansion is only lossless under specific conditions (e.g., $\sigma(I\sigma(x))=\sigma(x), \forall x$). Moreover, Net2Net struggles with 'weight symmetry' issues in the absence of noise addition, losing its lossless characteristic when noise is introduced.
> - As neural network architectures have evolved, particularly with the emergence of Transformers using LayerNorm, the constraints of Net2Net have become more pronounced. These architectures present challenges that Net2Net was not designed to address. Recognizing this gap, we introduce a suite of lossless method with **'weight symmetry breaking'**, applicable to both **divisible and indivisible width**, **depth expansion**, and across **various architectures** (See Table 1). The ability to maintain losslessness in these scenarios marks a substantial advancement.
>
>
> **Training recipe:**
> - Another significant advancement is our exploration of the impact of training recipes on the performance of expanded models. Net2Net does not explore this aspect, whereas our work provides novel insights into optimizing training strategies post-expansion, contributing to the enhanced performance of the expanded models.
>
> These novel contributions significantly address the critical limitations of existing model expansion techniques and may pave the way for new research and practical applications. The technical contribution and the practical applicability of our work align well with the standards of a top-tier conference.

---

### Author Response · Authors · 2023-11-17
**General response to all reviewers**

We express our sincere gratitude to all the reviewers for their valuable and constructive feedback on our manuscript. We are particularly thankful for the recognition of our work being **well-motivated** (reviewer u5r2), **well-organized** (reviewer u5r2), **versatile** (reviewers JSXE, pAyq, XnGd), **comprehensive** (reviewers JSXE, XnGd), and **insightful** (reviewers JSXE, XnGd). In response to the insightful comments and suggestions from the reviewers, we have made several modifications to our manuscript accordingly.

**Major modifications:**
1. Responding to JSXE's request, we conducted additional experiments with CNNs (ResNet-50 to WideResNet-110), detailed in Appendix F.2. LEMON demonstrated its effectiveness by outperforming bert2BERT-AKI and achieving a 33.3% reduction in computational resources compared to training from scratch.
2. Following pAyq's suggestion, we added experiments involving Post-LN Bert (BERT(6,384) to BERT(12,768)) in Appendix F.3. LEMON outperforms bert2BERT-AKI and achieves 36.57\% computational savings compared to training from scratch.
3. Addressing u5r2's feedback, we have clarified the distinction between 'weight symmetry' and 'breaking weight symmetry', highlighted in blue font color.
4. Addressing u5r2's feedback, we revised Table 2 to include the total training steps for each individual setup, providing a clearer comparison between different training methods.

**Others:**

5. In Appendix A, we added detailed information on model architecture and training configurations for better reproducibility.
6. Appendices D.3, D.4, and E.2 now contain details on applying LEMON to transformers with RMS Norm and CNN architectures, illustrating LEMON’s versatility.

---

### Public Comment · ~NAN_SHAO2 · 2023-12-06
**I am very interested in this paper.**

Thank you for the contributions made by the authors in this paper.
I'm a researcher deeply interested in this field. After reading this paper, I have some queries regarding the issues raised in the paper.

**Q1: As far as I know, there are several existing works highly related to this work. However, this paper dont compare with these methods.**

e.g.

**[1] Staged Training for Transformer Language Models**

[2] Learning to Grow Pretrained Models for Efficient Transformer Training

[3] 2x Faster Language Model Pre-training via Masked Structural Growth

I acknowledge that the method proposed in this paper appears to be more flexible compared to the methods in the aforementioned papers. But, it would be preferable to conduct detailed comparative experiments with existing works.

**Q2: From the perspective of continual pre-training, does the method proposed in this paper also ensure the 'training dynamics' property mentioned in paper [1]?**

I am looking forward to the authors response these two questions. I like this paper and it inspires me a lot.

---

### Meta-Review · Area_Chair_5jx5 · 2023-12-17

**Metareview:**

This paper presents a method for expanding model size during training, initializing a larger model from a pre-trained smaller model; experiments demonstrate the proposed strategy saves compute effort in comparison to training from scratch.  After the author response and discussion, all reviewers favor accept.  The AC agrees with the reviewer consensus.

**Justification For Why Not Higher Score:**

The overall practical impact (moderate training speedup) and spectrum of existing work in the area make the degree of contribution appropriate for acceptance as a poster.

**Justification For Why Not Lower Score:**

The paper adds a new method, with some attractive properties, to an extensive existing collection of techniques for model expansion.

---

### Decision · Program_Chairs · 2024-01-16

Accept (poster)